# Bayesian Social Deduction with Graph-Informed Language Models

## Abstract

Social reasoning—inferring unobservable beliefs and intentions from partial observations of other agents—remains a challenging task for large language models (LLMs). We evaluate the limits of current reasoning language models in the social deduction game *Avalon* and find that while the largest models demonstrate strong performance, they require extensive test-time inference and degrade sharply when distilled to smaller, real-time-capable variants. To address this, we introduce a hybrid reasoning framework that externalizes belief inference to a structured probabilistic model, while using an LLM for language understanding and interaction. Our approach achieves competitive performance with much larger models in Agent-Agent play and, notably, is the first language agent to defeat human players in a controlled study—achieving a 67% win rate and receiving higher qualitative ratings than both reasoning baselines and human teammates. We release code, models, and a dataset to support future work on social reasoning in LLM agents.

## 1 Introduction

Large language models (LLMs) have demonstrated remarkable general-purpose reasoning capabilities across a wide range of tasks (Ahn et al., 2024; Duan et al., 2024; Qiao et al., 2023), yet their ability to engage in social reasoning—particularly in multi-agent settings where participants hold private beliefs and (potentially) deceptive intentions—remains an open challenge (Mireshghallah et al., 2024; Shapira et al., 2024; Ullman, 2023). Recent studies (Li et al., 2023; Liu et al., 2024; Stepputtis et al., 2023) suggest that state-of-the-art LLMs often struggle to infer the latent goals and beliefs of other agents in such scenarios, limiting their effectiveness in settings that require theory of mind or strategic social deduction.

We revisit this problem in the context of *Avalon*, a social deduction game[1] that provides a structured yet complex environment for evaluating an agent's ability to infer hidden roles, manage uncertainty, and interact cooperatively or competitively with others by utilizing deception and persuasion. *Avalon* is particularly challenging as it requires agents to utilize *constrained probabilistic reasoning* over long temporal horizons, an aspect rarely seen in prior benchmarks. Consider the following example.

> **Prompt**: *There are five players (Alice, Bob, Carol, Dave, and Eve), two of which are Evil while the rest are Good. The first party (consisting of Alice and Bob) and the second party (Carol and Dave) both failed. If each Evil player has a 70% chance to fail the quest, what is the probability that each player is Evil?*
>
> **LLM:** *Each individual appears in exactly half of the possible pairs. Therefore, the probability that any specific individual is Evil is: 0.5*

Despite its simplicity, state-of-the-art large reasoning models (LRMs) (Xu et al., 2025; Zhou et al., 2025), including 8B and 70B variants of Deepseek-R1, fail to successfully reason that the only player to not appear in a party, Eve, has a 0% probability of being Evil, as there are only two Evil players. While larger models are capable of solving this trivial example, they too struggle as the temporal horizon increases and social aspects are brought into play. Furthermore, performance gains come at

---

[1]Although referred to as "social *deduction*", reasoning in social deduction games is inherently probabilistic—due to epistemic uncertainty—rather than purely deductive (logical or mathematical).

a significant computational cost, requiring long chains of reasoning tokens, rendering such models impractical for real-time interactive play with human users.

**To overcome this limitation, we propose a *hybrid reasoning framework* that augments LLMs with structured probabilistic inference over beliefs**, combining the linguistic grounding and rich priors of foundation models with the rigor of Bayesian reasoning. In this paper, we present **GRAIL (Graph Reasoning Agent Informed through Language)**, a hybrid framework where an LLM handles dialogue parsing, generates utterances, and interprets informal social cues, while a probabilistic graphical model tracks latent roles and beliefs by analyzing observed game events and social interactions. This decoupling makes belief inference both interpretable and efficient, avoiding the need for extensive token generation during gameplay.

Despite using a significantly smaller LLM, our method matches or exceeds the performance of large-scale reasoning models across multiple metrics, including win rate, belief accuracy, and belief consistency in Agent-Agent *Avalon* games. Notably, GRAIL is, to the best of our knowledge, the first language agent to successfully play and win against novice human players in a controlled participant study, *achieving a striking 67% win rate*. In post-game surveys, participants rated GRAIL's contributions and helpfulness significantly higher than reasoning model baselines, and in many cases, even over other human players. These results suggest that external structured reasoning models effectively complement LLMs, enabling socially competent behavior in real-time interactive settings.

To support future work on social reasoning in multi-agent environments, we release our framework, agent implementations, a new benchmark, and a dataset of Agent-Agent and Human-Agent *Avalon* games, which include player discussions and associated game states. Together, these contributions provide a testbed for studying social inference, deception, and cooperation in LLM agents.

## 2 BACKGROUND AND RELATED WORK

**Social Deduction Agents:** Social deduction games provide a natural testbed for evaluating the social reasoning capabilities of LLMs (Lan et al., 2024). Previous studies have applied LLMs to hidden-role and social deduction games such as *Werewolf* (Lai et al., 2023; Wu et al., 2024; Xu et al., 2024b;c), *Among Us* (Sarkar et al., 2025), *Avalon* (Light et al., 2023; Wang et al., 2023), and *Mafia* (Ibraheem et al., 2022). Before the advent of foundation models, the DeepRole agent (Serrino et al., 2019) was trained via self-play to play 5-player versions of *Avalon* without natural dialogue. More recently, Stepputtis et al. (2023) explored the use of LLMs for hidden role inference based on long-form dialogue in *Avalon* games and demonstrated their shortcomings. In parallel, probabilistic graphical models have also been explored in the context of social deduction games (Xu et al., 2024a).

**Theory of Mind:** Theory of mind (ToM), the ability to attribute mental states like beliefs, desires, and intentions to oneself and others (Ho et al., 2022), is crucial for social reasoning, especially for deception and persuasion (Alon et al., 2023; Ding et al., 2015). The presence of ToM-like abilities in Large Language Models (LLMs) is currently debated (Kosinski, 2024; Shapira et al., 2024; Strachan et al., 2024). Notably, Riemer et al. (2025) argue that high performance on ToM benchmarks may not reflect genuine ToM reasoning in LLMs, as these abilities may not extend to novel scenarios. Social deduction games offer a robust environment for probing these limits, as success requires agents to model and reason about the intentions, beliefs, and likely actions of others (Guo et al., 2024; Zhou et al., 2023; , FAIR). More recently, Sclar et al. (2023) showed that graph-based models, combined with LLMs, can support belief reasoning in standard ToM tasks.

**Scaling and Reasoning:** LLMs exhibit *discontinuous* improvements in zero-shot reasoning with increased model size (Chowdhery et al., 2023; Wei et al., 2022), a trend that extends to common-sense and social reasoning (Shapira et al., 2024). Besides scaling parameters, reasoning ability can be enhanced by scaling test-time token generation (Zhang et al., 2025). Recent work has posited that these reasoning models are capable of step-by-step problem-solving on a variety of benchmarks, even with fewer parameters (DeepSeek-AI, 2025; Team, 2025; Zhang et al., 2025). This presents a trade-off between *test-time computation* and *parameter count* as an alternative means for improving social reasoning capabilities (Muennighoff et al., 2025; Snell et al., 2024). However, several recent studies argue that benchmark gains alone are not evidence of emergent reasoning capabilities (Schaeffer et al., 2023; Yu et al., 2025).

Figure 1: Overview of GRAIL's architecture and inter-module communication. A factor graph tracks beliefs over hidden player roles using belief propagation, informed by game-state observations and an LLM-generated language prior. Conditional probabilities are estimated by a neural network trained on historical games. Inferred beliefs guide both action selection and message generation.

**The Resistance *Avalon*:** In the social deduction game *Avalon* (Eskridge, 2012), players belong to either the Good team (who try to complete quests) or the Evil team (who try to sabotage them). The game consists of five rounds with quest parties of 2, 3, 4, 3, and 4 members, respectively. Each round, players propose and vote on a party. If approved, its members secretly vote on the quest's outcome; the quest succeeds only if all members vote success. Good wins by completing three quests; Evil wins by failing three. Players communicate via turn-based chat (see Appendix A for full details). We develop an AI agent that plays exclusively as Good, such that we can focus on identifying Evil opponents.[2]

## 3  BAYESIAN BELIEF INFERENCE WITH FACTOR GRAPHS

To identify Evil players in *Avalon*, an agent must be able to form and support parties composed entirely of Good players, despite lacking knowledge of other players' roles. This is a **constrained probabilistic reasoning** task, where agents infer latent player roles from observable actions and unstructured natural language dialogue, and then act based on the certainty of their beliefs. This is a challenging task for state-of-the-art language models that rely purely on token-level reasoning. We introduce a hybrid approach that externalizes inference to a structured graphical model well-suited to constrained reasoning. The language model attends to social-linguistic signals while the reasoning model maintains and updates beliefs, enabling strong performance even with small models. Our approach is designed around two core objectives.

**Constraint Satisfaction:** Deduction in *Avalon* depends on satisfying a combination of *hard* and *soft* constraints. For example, the fixed number of Evil players (e.g., two) imposes a hard constraint on valid role assignments. Similarly, a failed quest implies at least one Evil member in the party, introducing a soft constraint that influences belief updates. However, many possible role assignments satisfy these constraints, and agents must consider multiple plausible hypotheses simultaneously.

**Probabilistic Inference:** To support reasoning about plausible role assignments, we model player roles and relevant game variables as random variables in a probabilistic model, allowing us to represent uncertainty over role assignments and update beliefs as new evidence accumulates. We formulate hidden role inference as probabilistic inference over a *factor graph*, which compactly models dependencies and enforces game-specific constraints.

### 3.1  FACTOR GRAPHS FOR SOCIAL DEDUCTION

A factor graph is a bipartite graph, defined as the triplet $\mathcal{G} = (\mathcal{V}, \mathcal{F}, \mathcal{E})$ where $\mathcal{V} = \{X_1, X_2, \ldots, X_n\}$ is the set of *variable nodes*, $\mathcal{F} = \{f_1, f_2, \ldots, f_m\}$ is the set of *factor nodes*, and $\mathcal{E} \subseteq \mathcal{V} \times \mathcal{F}$ is the set of *edges*. Each factor represents a function: an edge $(X_i, f_j) \in \mathcal{E}$ exists if and only if $X_i$ is an argument of $f_j$. A factor function can represent a probability distribution if it is normalized, or a constraint if its values are either 0 or 1. Through this, the graph is able to represent dependencies and constraints between a set of variables.

The variable nodes in our factor graph represent both game state and player role variables, with the goal of identifying which players are likely to be Evil. The role variables are denoted

---

[2]We exclude special roles, e.g. Merlin, to focus on detecting deception rather than producing it.

$\mathcal{R} = \{r_1, \ldots, r_6\}$, where $r_j \in 0, 1$ indicates whether player $j$ is Good (0) or Evil (1). The game state variables are given by $\mathcal{S} = \{p_1, v_1, o_1 \ldots, p_6, v_6, o_6\}$, where $p_i, v_i, o_i$ represent the party composition, voting outcome, and quest result for quest $i$, respectively. A binary factor enforces the hard constraint that exactly two players are Evil. For each role variable $r_j$, we include a factor connected to the game state that encodes its conditional probability, defined as $F = p(r_j | \{p_i, v_i, o_i | \forall i\})$. An overview of the factor graph structure is shown in Fig. 1, with details provided in Appendix B.

We use max-product belief propagation (Wainwright & Jordan, 2008) to perform approximate maximum a posteriori (MAP) inference over the factor graph, identifying the most likely assignment of hidden roles given the observed game state. Unlike the more common sum-product algorithm, max-product directly estimates a MAP assignment rather than integrating over alternatives (Kschischang et al., 2001; Murphy et al., 1999) and is better suited for handling deterministic constraints (Smith & Gogate, 2014), e.g., the number of Evil players. In our setting, the agent's role is known, so inference is performed over the remaining five hidden role variables. The max-product algorithm calculates the max-marginals of each hidden variable, which is proportional to *the probability of player $i$ being Evil* and can be treated as the belief about that variable(Yanover & Weiss, 2003). From this point forward, we refer to this max-marginal as the *belief* of the agent about the player $i$, $b_i$. The details of the max-product algorithm and its scalability can be found in Appendix B.2, B.7.

### 3.2 FACTOR FUNCTION APPROXIMATION

Traditionally, factor functions are represented as probability tables, which are impractical in high-dimensional settings such as ours. To address this, we approximate the conditional probability distribution in each factor using a simple feedforward neural network (Richard & Lippmann, 1991). Each factor corresponds to the conditional density $p(r_j | \{p_i, v_i, o_i | \forall i\})$, where $r_j$ is a binary role variable. We model this as a binary classification task, using a sigmoid output to estimate the conditional probability of $r_j$. The network is trained on a dataset of over 100,000 games[3] – consisting only of game states without language – collected from AvalonLogs[4] and ProAvalon.[5] To account for temporal partial observability, we apply a masking scheme that zeroes out future inputs, ensuring the model only conditions on information that would have been available at a given point in the game. Details on the network architecture and training procedure are provided in Appendix B.3.1, B.4.

**Mitigating Positional Bias in Factor Functions:** In *Avalon*, players take turns according to a fixed sequence, which introduces positional bias during training and inference. To mitigate this, we augment the training data with all circular permutations of player orderings. Additionally, using separate neural networks for each factor node can also introduce positional bias, which we avoid by using a shared factor function across all nodes. We apply an ego-centric transformation to the input state such that the player corresponding to the current factor is always placed in the first index while preserving the relative positions of other players.

## 4 GRAIL: GRAPH REASONING AGENT INFORMED THROUGH LANGUAGE

Fundamentally, GRAIL is a hybrid model composed of three interconnected components: an LLM, a factor graph for tracking beliefs over player roles, and a heuristic action policy, as illustrated in Fig. 1. The factor graph maintains and updates probabilistic beliefs about player roles based on observable game events. These beliefs are then passed to the LLM, which generates contextually appropriate dialogue based on the agent's current understanding of the game.

**Game Actions:** Proposing parties and voting for them follows a heuristic policy derived from factor graph beliefs: a party is proposed or approved only if all members are more likely Good than Evil. Furthermore, when our GRAIL agent is the party leader, the heuristic guides the agent through the necessary stages of proposing a party, initiating a discussion, adjusting the proposal if necessary, and initiating a party vote. For more details refer to Appendix E

---

[3]In Appendix B.5 we show that only 2.5K-5K games are required for sufficient predictive performance.
[4]https://github.com/WhoaWhoa/avalonlogs
[5]https://www.proavalon.com

Table 1: Win rates across different team compositions. Each matchup consists of 20 games.

(a) Win rates for homogeneous agent teams.

| Good Team | Evil Team | | | | Avg |
|---|---|---|---|---|---|
| | Rand | ReCon | DS-R1 | o4-mini | |
| Rand | 0.00 | 0.00 | 0.00 | 0.00 | 0.00 |
| DeepSeek-R1 | 0.90 | 0.35 | **0.70** | **0.90** | 0.71 |
| GPT-o4-mini | 0.70 | 0.05 | 0.25 | 0.50 | 0.40 |
| ReCon | 0.80 | 0.15 | 0.50 | 0.25 | 0.43 |
| **GRAIL** | **0.95** | **0.45** | **0.70** | **0.90** | **0.75** |

(b) Win rates for mixed GRAIL/ReCon teams.

| Good Team Comp. | Evil Team | | Avg |
|---|---|---|---|
| | ReCon | o4-mini | |
| 0 GRAIL & 4 ReCon | 0.15 | 0.25 | 0.20 |
| 1 GRAIL & 3 ReCon | 0.20 | 0.25 | 0.23 |
| 2 GRAIL & 2 ReCon | 0.20 | 0.65 | 0.43 |
| 3 GRAIL & 1 ReCon | 0.40 | **0.90** | 0.65 |
| 4 GRAIL & 0 ReCon | **0.45** | **0.90** | **0.68** |

### 4.1 Incorporating Language Priors into Belief Propagation

A core part of *Avalon* is the dialogue between players, providing valuable insights into whether or not a player is Good or Evil. During the discussion of parties and quests, players may contradict themselves, hint at alliances, or reveal privileged knowledge. To incorporate such information, we utilize an LLM to estimate priors for beliefs over player roles, which are subsequently integrated into max-product belief propagation for downstream reasoning.

Formally, for player $j$ we define a prior probability $p(r_j^t)$ over their role at time step $t$, where $r_j^t = 1$ indicates that player $j$ is Evil. By default, this prior is uninformative, i.e., uniform, but we use the LLM to adjust this prior and incorporate language feedback. We present the LLM with the current chat history and the belief of player $j$, $b_j^{t-1}$, and ask it to assess whether the belief should be *higher*, *lower*, or remain the *same*. The LLM's qualitative judgement $\delta_j^t \in \{\texttt{higher}, \texttt{lower}, \texttt{same}\}$ is converted into a numeric prior using a pre-defined mapping parameter $\beta^t$ as follows:

$$p(r_j^t) = \begin{cases} 0.5 + \beta^t & \text{if } \delta_j^t = \texttt{higher} \\ 0.5 - \beta^t & \text{if } \delta_j^t = \texttt{lower} \\ 0.5 & \text{if } \delta_j^t = \texttt{same}. \end{cases}$$

In practice, we treat $\beta$ as a tunable hyperparameter. To avoid overconfidence in early rounds when little evidence is available, we use values close to $0$ and increase them as the game progresses.

We adopt this qualitative prompting scheme instead of directly asking LLMs to generate probabilities because LLMs often struggle to accurately interpret, manipulate, and generate numeric data (Schwartz et al., 2024; Yang et al., 2025). All prompt templates used to extract priors and generate messages are provided in Appendix F.

## 5 Experiments

To evaluate GRAIL, we simulated games against synthetic Evil players and assessed its performance across a wide range of metrics, including win rate, belief accuracy, and belief consistency.

**Baselines:** We compared GRAIL against reasoning and non-reasoning agents. Reasoning agents use LRMs for both action selection and message generation, including **DeepSeek-R1** (DeepSeek-AI, 2025) and OpenAI's **GPT-o4-mini** (OpenAI, 2025). Non-reasoning agents use LLMs but may still employ manual chain-of-thought where applicable, such as **ReCon** (Wang et al., 2023). Finally, a **Random** agent serves as a performance lower bound.

**ReCon:** Recursive Contemplation (ReCon), is a cognitive framework for LLM agents inspired by human recursive thinking. ReCon acts in two stages: "Formulation Contemplation" for initial internal thought and first-order perspective-taking, and "Refinement Contemplation" for re-evaluating and refining speech with second-order perspective-taking. Each one of these stages are implemented through extensive prompt engineering.

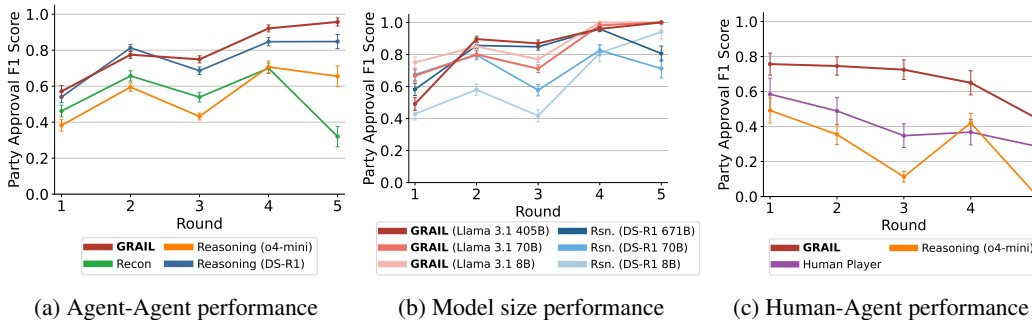

(a) Agent-Agent performance     (b) Model size performance     (c) Human-Agent performance

Figure 2: F1 scores of agents' voting predictions of team composition per round (error bars indicate SE) (a) GRAIL compared to other baseline agents, (b) ablation of GRAIL on non-reasoning Llama 3.1 model compared to DeepSeek-R1 reasoning model across different parameter sizes, (c) GRAIL compared to human players and reasoning models used in the human study.

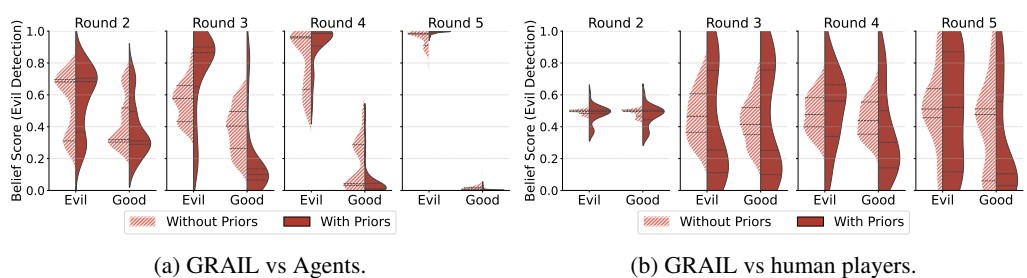

(a) GRAIL vs Agents.        (b) GRAIL vs human players.

Figure 3: Probability density of GRAIL beliefs about Good and Evil players, with and without using LLM prior in 5-round games (a) 13 games against ReCon and reasoning agents (DS-R1, GPT-o4-mini) (b) 4 games against human players.

## 5.1 AGENT-AGENT EVALUATION

To systematically evaluate agent performance, we constructed a four Good agent vs. two Evil agent matchup matrix. Each pairing was tested over 20 games, with both GRAIL and ReCon utilizing GPT-4.1 as the underlying LLM. The results (Table 1a) show that **GRAIL achieves the highest win rate (average of 75%) among all Good agents**, consistently outperforming both reasoning and non-reasoning baselines, including those using the 671B DeepSeek-R1 LRM.

We further analyzed each agent's votes with respect to proposed parties, treating these votes as binary predictions of whether a party contains an Evil player. Fig. 2a compares the F1 scores of these predictions across game rounds, showing that GRAIL again outperforms all baselines. This result suggests that GRAIL is particularly effective at reasoning over long horizons, as reflected by strong late-game performance (after the third round). In contrast, GPT-o4-mini and ReCon exhibit a performance drop in the fifth round when the context horizon is the longest.

**Token Analysis:** To evaluate agent efficiency, we computed the average number of input and output tokens per round. Input tokens reflect the amount of context and guidance provided, while output tokens capture the length of the reasoning chain and implicitly indicate relative compute costs. The results are shown in Fig. 4, where we see that **GRAIL produces more than 10 times fewer output tokens than all other baselines**, underscoring the computational efficiency of our method. Notably, unlike the LLM-based ReCon agent, which requires multiple prompts per turn as part of its reasoning process, GRAIL completes reasoning in a single prompt, resulting in far fewer input tokens.

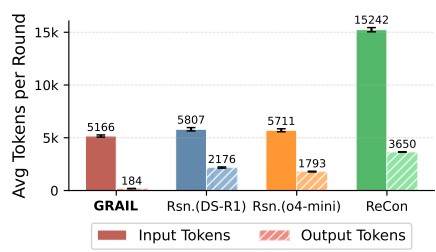

Figure 4: Average per-round token usage for GRAIL, LRM-based reasoning agents, and ReCon in Agent-Agent games.

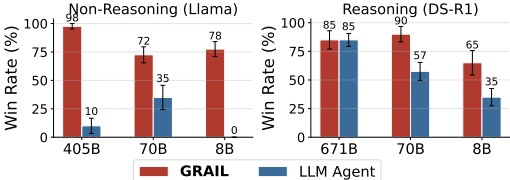

(a) Architecture ablations for GRAIL in which we use only belief propagation (Graph Only) or language priors (LLM Only) using Llama 3.1 with different sizes (405B, 70B, and 8B parameters) over 40 games.

(b) Win rates for GRAIL and the LLM-based agent for different underlying models across varying parameter scales. (Left) Llama 3.1 non-reasoning models and (Right) DeepSeek-R1 reasoning models.

Figure 5: Combined ablation results across agent components, model size, and reasoning types.

**Mixed-Team Setting:** In the mixed-team setting, we tested GRAIL's effectiveness when paired with weaker agents, such as ReCon, as teammates. We observed that gradually adding GRAIL agents to the team immediately improves performance (see Table 1b), depending on the opposing team's capabilities. These findings underscore GRAIL's ability to improve overall team performance, even when partnered with less capable agents.

**Belief Distribution:** To analyze the effect of language priors, we visualized the evolution of GRAIL's belief over the course of 13 games that ended after exactly 5 rounds. Fig. 3a shows the kernel density estimations $\text{KDE}(b_j^t \mid r_j = 1)$ (Evil player) and $\text{KDE}(b_j^t \mid r_j = 0)$ (Good player), computed both *with* and *without* the prior $p(r_j^{t-1})$ in agent–agent games. We observe that beliefs about both Good and Evil players progressively converge toward their true values as the game advances. Early-game distributions are uncertain, whereas late-game distributions exhibit high-confidence peaks. Incorporating the language prior accelerates this convergence, resulting in confident and accurate beliefs by round three, compared to rounds four or five without the prior.

## 5.2 MODEL SIZE AND ARCHITECTURE EVALUATION

To evaluate the contribution of GRAIL's individual components, we performed ablation studies under two conditions: **LLM Only**, in which beliefs are set directly from the prior ($b_j^t = p(r_j^t)$), and **Graph Only**, which uses belief propagation without the language prior (by fixing $\beta^t = 0$). Furthermore, to understand how these design choices impact sensitivity to LLM size, we pair this analysis with an ablation study on the size of the underlying LLM, using the Llama 3.1 family (Dubey et al., 2024) with 8B, 70B, and 405B parameters for GRAIL.

Ablation results (Fig. 5a) demonstrate that the full GRAIL method, combining both the factor graph and language priors, consistently outperforms both ablated variants. The LLM Only variant is highly sensitive to model size, exhibiting sharp performance degradation with smaller models. Conversely, the Graph Only variant is robust to LLM size and maintains a high win rate (75%) even with the smallest 8B model. From this, we conclude that **the factor graph establishes a "performance floor," effectively mitigating the negative performance impacts of smaller models**.

We next evaluated the sensitivity of reasoning agent performance to model size by playing games where the original 671B DeepSeek-R1 model is replaced with smaller, distilled variants (70B and 8B). This allows us to directly measure how reasoning quality degrades with reduced model capacity. To further isolate the effect of reasoning ability, we also substituted the reasoning agent's LRM with comparably-sized Llama LLM models; similarly, we evaluated GRAIL with DeepSeek LRMs.

The results (Fig. 5b) highlight GRAIL's robustness to model size, which is a sharp contrast to the reasoning agents. Specifically, the reasoning agents exhibit poor performance when using smaller LRMs or non-reasoning LLMs. This results in two key insights: 1) the LLM-based GRAIL outperforms similarly-sized LRM-based reasoning agents in every size class, and 2) GRAIL achieves *higher* win rates using a smaller LLM than reasoning agents using much larger LRMs, e.g., GRAIL 8B Llama outperforms reasoning 70B DeepSeek-R1. We also observed a counterintuitive result: the win rate of the reasoning agent with the 405B Llama model is *worse* than the 70B Llama model. Upon analysis of chat messages, we observed that this is due to high sycophancy (Sharma et al., 2024) as the Good agents complied with the Evil agents' requests, e.g. "I should be in the party."

Table 2: Average per-turn wall-clock time (in seconds) of agents over different methods and model sizes. Times for GRAIL include graph inference.

|  | 8B | 70B* | 405B / 671B |
|---|---|---|---|
| DS-R1 (s) | 17.37±20.59 | 15.01±6.55 | 85.50±179.29 |
| GRAIL (s) | 14.04±2.00 | 18.73±1.82* | 20.00±9.99 |
| Graph (s) | 5.05 | 10.15* | 5.23 |

Table 3: Belief F1 score of GRAIL using the priors generated by different Llama model sizes with different $\beta$ values.

| Model Size | $\beta$ | | | | |
|---|---|---|---|---|---|
|  | 0.05 | 0.10 | 0.15 | 0.20 | 0.25 |
| 8B LLaMA | **0.89** | 0.82 | 0.79 | 0.78 | 0.76 |
| 405B LLaMA | 0.88 | 0.89 | **0.90** | 0.89 | 0.89 |

**Voting Dynamics:** A comparison of round-by-round voting patterns shown in Fig. 2b highlights that GRAIL yields higher F1 scores than the DeepSeek-based reasoning agent at comparable parameter scales (Llama 3.1 405B vs. DeepSeek 671B; 70B vs 70B; 8B vs 8B) Our agent demonstrates greater consistency and reduced performance degradation across all model sizes.

**Time Analysis:** We compared GRAIL and the best reasoning agent (DeepSeek-R1) on average time per turn across model sizes, noting hardware differences[6]. For GRAIL, we separately measured the graph propagation time in addition to the total turn time [7]. DeepSeek-R1 shows a high variance due to time spent on internal reasoning, while GRAIL is consistently faster (see Table 2).

**The Effect of Beta:** To evaluate the effect of the $\beta$ parameter (from section 4.1), we reran the belief propagation on games played by the 8B and 405B agents (Appendix C). We find that priors from smaller models produce more accurate beliefs (F1 score) when paired with a smaller $\beta$ (Table 3). This suggests smaller models generate lower-quality priors. To verify this, we measured the priors' performance on classifying Evil players and found that the 405B model (0.73 F1) significantly outperforms the 70B (0.60 F1) and 8B (0.47 F1) models. **This confirms that smaller models struggle to detect deception**, validating our approach of tuning $\beta$ as a model-specific hyperparameter.

**Hallucination Analysis:** We measured agent alignment with the game state by analyzing message hallucination rates for GRAIL and the reasoning agent across various model sizes. We utilized GPT-4.1 as a judge to detect hallucinations (Gu et al., 2025), as it has proven to agree with human judgment in 95% of the cases (see Appendix G). Across all sizes, GRAIL consistently hallucinates less than the reasoning agent (Fig 6), indicating stronger grounded reasoning through our hybrid approach. We observed that the reasoning agent tends to make speculative statements, a behavior that could impair trust and coordination in Human-Agent games.

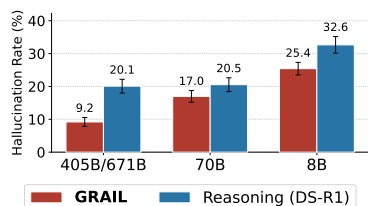

Figure 6: Hallucination rates for GRAIL (Llama 3.1) and the reasoning agent (DS-R1) for different model sizes over 40 games.

## 6 HUMAN EVALUATION

Lastly, we tested GRAIL's ability to play with and against human players, demonstrating GRAIL's ability to handle diverse strategies that create dynamic and unpredictable gameplay. To this end, we conducted an IRB-approved within-subjects study spanning 44 participants where agents played against Evil human players alongside one Good human teammate in real time. After each game, participants completed questionnaires on the perceived contribution and helpfulness of Good players, without knowing the study's purpose to reduce bias. Additional methods appear in Appendix I.

The following hypotheses guide our evaluation: **H1:** Good teams consisting of GRAIL agents will win significantly more games than those composed of reasoning agents. **H2:** GRAIL will identify Evil players by rejecting proposals containing Evil players and accepting proposals exclusively composed of Good players more accurately, compared to the reasoning agents (**H2.1**) and human players (**H2.2**). **H3:** Participants will state that GRAIL contributed to the success of the Good team more, compared to the reasoning agents (**H3.1**) and human players (**H3.2**). **H4:** Participants will

---

[6]The 70B GRAIL ran on a different hardware with a weaker CPU compared to the 8B and 405B variants
[7]With GPU-optimized belief propagation, GRAIL could be even more efficient.

prefer the helpfulness of the suggestions and comments of GRAIL more, compared to the reasoning agents (**H4.1**) and human players (**H4.2**).

**Setup:** Each experiment paired two human players as Evil with one human as Good, alongside three Good agent teammates. Every participant played two games, alternating between GRAIL and baseline reasoning agents (GPT-o4-mini) in randomized order for fairness. Participants were unaware of the presence of non-human players. Due to latency and reliability issues with the DeepSeek-R1 API, GPT-o4-mini was used as the baseline model.

This configuration enables evaluations from two perspectives: (1) Evil players interacting **against** agents, and (2) a Good player collaborating **alongside** agents. After each game, human evaluators answered two questions assessing the overall contribution and communication quality of the Good team players on a five-point Likert response scale. These questions are:
Q1: "Player _ contributed to the success of the Good team."
Q2: "Player _ made suggestions or comments that were helpful to the Good team."

### 6.1 HUMAN STUDY EVALUATION

We conducted 15 trials with three participants each, running 15 games with GRAIL and 15 with the reasoning agent. Fig. 7 presents the results of the qualitative evaluation. Because participants were unaware that both humans and agents were present on the Good team, Evil players evaluated all Good players, without distinguishing between agents and humans.

**H1:** Across 15 games, GRAIL won 10 and lost 5 (67% win rate), whereas the reasoning-based agent won 4 and lost 11 (27% win rate). We assessed the statistical significance of this performance difference. A frequentist test yields p = 0.054, just above the 0.05 threshold, but suggests a favorable trend. Bayesian analysis produces a 96.7% posterior probability that GRAIL outperforms, with a 95% credible interval of (0.482, 0.939), leaving only a 1.8% chance the reasoning agent is superior.

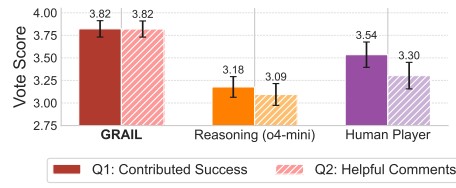

Figure 7: Average scores given to agents by humans across two questions assessing contribution and helpfulness. Human ratings (Evil players' votes for Good human players) are included for baseline comparison.

**H2:** We evaluated predictive performance for party proposal assessments across human study games. Figure 2c shows GRAIL consistently votes more accurately than human players. A one-tailed Wilcoxon (1945) signed-rank test of F1 scores across 15 games confirms GRAIL's statistically significant improvement over humans ($p = 0.007$, **H2.1**). A Mann-Whitney U test (Mann & Whitney, 1947) comparing GRAIL and reasoning agents' F1 score distributions across the same games further supports this finding ($p = 0.0103$, **H2.2**).

**H3, H4:** We conducted one-tailed t-tests (p < 0.05) comparing GRAIL, reasoning agents, and human players on both questions. GRAIL significantly outperformed reasoning agents in task contribution (Q1: $p = 0.001$, **H3.1**) and suggestion quality (Q2: $p = 0.0005$, **H4.1**). Compared to humans, GRAIL showed a non-significant trend in task contribution (Q1: $p = 0.105$, **H3.2**) but significantly better suggestions (Q2: $p = 0.035$, **H4.2**). Results indicate GRAIL outperforms reasoning agents and approaches human-level performance in effectiveness and helpfulness.

## 7 CONCLUSION

In this work, we propose GRAIL, a novel approach to hybrid reasoning that utilizes a structured probabilistic inference framework to identify and track player roles in a complex and challenging social deduction game – *Avalon* – that requires constrained probabilistic reasoning. Through our extensive experiments, we demonstrate that current state-of-the-art reasoning models struggle in such settings, underlining the benefit of our proposed hybrid model, significantly outperforming prior work and LRMs while using much smaller non-reasoning LLMs. Furthermore, we demonstrate that GRAIL is capable of playing with and against human players, achieving a win rate of 67% against novice human players. In its current state, GRAIL is exclusively designed as a Good agent for de-

tecting rather than generating deception, using first-order Theory of Mind through Bayesian reasoning over a factor graph, complementary to the LLM inference. Generating deception or persuasion (e.g., required to successfully play special roles such as Merlin) requires second-order reasoning, which builds on our strong first-order foundation. In future work, we will extend GRAIL to model and utilize second-order beliefs via conditional probability, enabling the detection of more intricate deception as well as improving GRAIL's persuasiveness.

ETHICS STATEMENT:

The paper emphasizes technical contributions and has little to no potential positive or negative societal impacts. Although the results of the study can imply a potential positive impact, we do not mention this in the paper, nor do we explore any potential negative impacts. The human-subject study is IRB-approved and conforms to the ICLR Code of Ethics. Before participating in our experiment, all participants signed a consent form disclosing all potential risks, and after the experiment, all participants signed the post-experiment deception form. All participants gave informed consent for us to use the data collected from them, and were compensated for their time.

REPRODUCIBILITY STATEMENT:

The paper describes the setups, datasets, underlying models, model sizes, hyperparameters, and evaluation metrics used in the experiments. Prompts and the questions asked of participants can be found in the Appendix. The anonymized human experiment results and the code are provided in the supplementary material.

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

# APPENDIX FOR BAYESIAN SOCIAL DEDUCTION WITH GRAPH-INFORMED LANGUAGE MODELS

## A  THE RESISTANCE: AVALON

The Resistance: Avalon is a standalone social deduction game designed by Don Eskridge and published by Indie Boards and Cards in 2012 Eskridge (2012). It builds upon the foundation laid by its predecessor, The Resistance, which Eskridge also designed and released in 2009. While both games share core mechanics involving hidden roles and team-based missions, Avalon introduces a rich Arthurian theme and additional character roles that deepen strategic play. Avalon is designed for 5-10 people, where players are split into two teams: Good (Loyal Servants of Arthur) and Evil (Minions of Mordred). In this paper, we focus on a simplified version with 6 players, which does not include special roles (e.g. Merlin, Assassin, etc.) to better focus on detecting deception, rather than producing it.

In the simplified version of the game with 6 players used in our study, there are 4 Good players and 2 Evil players. Roles are randomly assigned and kept secret. Each Evil player knows the identities of their Evil teammate, whereas Good players do not know the identities of any other player. The game progresses through 5 rounds, requiring parties of 2, 3, 4, 3, and 4 members, each consisting of a party proposal, a discussion phase, a party vote, and a quest vote. Players take turns in a clockwise direction, starting with the player designated as the leader. The leader proposes a team of a predetermined size to participate in the quest. Players then discuss the proposal in clockwise order. After everyone has had their turn to speak, there is a vote on whether to approve or reject the current party. If the party is approved by a majority, then the players proceed to the quest vote. If rejected, leadership passes on to the next player clockwise. If five consecutive parties are rejected, the Evil team wins by default. During the quest vote, party members secretly vote on the quest outcome, which succeeds only if players vote unanimously for success. The Good team wins if three missions succeed. The Evil team wins if three missions fail or if five consecutive team proposals are rejected.

## B  FACTOR GRAPH

### B.1  STRUCTURE

The detailed structure of the factor graph used in **GRAIL** is shown in Fig 8. In this visualization, we represent the variables with capital letters (as random variables). More importantly, in our implemented graph structure, we only consider the final approved party for each quest.

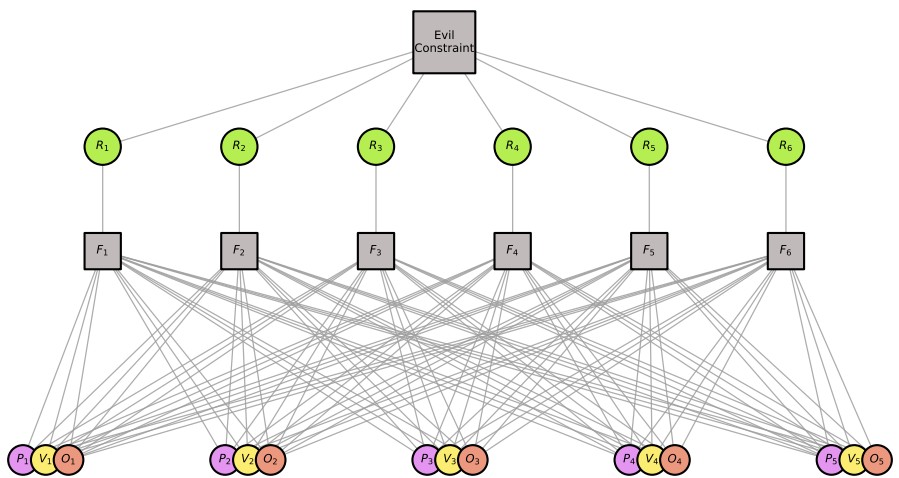

Figure 8: The Factor Graph Structure

$R_i$ is a binary random variable representing the role of the player $i$. If the player is Evil, $R_i = 1$, otherwise $R_i = 0$.

The factors can represent either a constraint (through a binary function) or a probability (throgh a joint or conditional probability function) depending on their purpose. The *Evil Constraint* enforces the constraint that only two of the players are Evil, and is defined as a function of $R_1 \ldots R_6$ as seen in Eq 1:

$$
f_{\text{Evil Constraint}}(R_1, \ldots, R_6) \;=\; \mathbf{1}_{\left\{ \sum_{i=1}^{6} R_i = 2 \right\}} \;=\; \delta\left( \sum_{i=1}^{6} R_i - 2 \right) \;=\; \begin{cases} 1, & \sum_{i=1}^{6} R_i = 2, \\ 0, & \text{otherwise.} \end{cases} \tag{1}
$$

$P_j$, $V_j$, and $O_j$ are categorical random variables representing the party at quest $j$, the public vote for the party in quest $j$, and the outcome of quest $j$, respectively. We use a simple numerical encoding to save and represent the party, vote, and outcome of each quest. We consider value zero to indicate *unseen* or future quests; in other words, $P_i = V_i = O_i = 0$ means that quest $i$ has not happened yet. In the upcoming sections, the indexes and encoding start from number 1 due to this consideration.

**Party ($P_j$):** Assuming that the party has $k$ members, we list all $k$-element subsets of the players in increasing lexicographic order called $S$. Given a party composition, the encoding will be the index of that party in the ordered list $S$. Thus, for example, $P_1 = 1$ encodes $\{1, 2\}$ (party with player 1 and 2), $P_1 = 2$ encodes $\{1, 3\}$ and so on. Based on this encoding, we will have:

$$
\begin{aligned}
P_1, \ P_3, \ P_5 &\in \{0, \ldots, 15\}, \\
P_2, \ P_4 &\in \{0, \ldots, 20\}.
\end{aligned}
$$

**Vote ($V_j$):** Since the parties are selected by a majority vote, the number of players on the list of approval votes will be either 4, 5, or 6, leading to a total of 22 possible vote compositions. Similar to the encoding used for the Party, we order these vote compositions (represented by subsets of the player list) in an increasing lexicographic order. The encoding of a party vote will be the index of that vote composition in this ordered list.

$$
V_i \in \{0, \ldots, 22\}
$$

**Outcome ($O_j$):** We encode success in quest $j$ as $O_j = 2$ and failure as $O_j = 1$.

**Factor Nodes:** The factor nodes represent the conditional probability of the role node given the state of the game. The details of approximating this conditional probability using a neural network are given in Section B.3.

### B.2  BELIEF PROPAGATION

The max-product belief propagation algorithm works by passing messages along the edges of the factor graph and updating them iteratively. These messages represent the "influence" that variables and factors have on each other in terms of maximizing the global function. There are two types of messages:

- **Message from variable $x_i$ to factor $f_a$:** This message represents the "belief" of variable $x_i$ about its state, based on information received from all other connected factor nodes *except $f_a$*. It is calculated as the product of all incoming messages to $x$ from neighboring factor nodes as seen in Eq 2, where $N(x_i)$ is the set of factor nodes connected to $x_i$:

$$
\mu_{x_i \to f_a}^{(t)}(x_i) = \prod_{f_b \in N(x_i)/\{f_a\}} \mu_{f_b \to x_i}^{(t-1)}(x_i) \tag{2}
$$

- **Message from variable $f_a$ to factor $x_i$:** This message represents the "belief" of factor $f_a$ about the state of variable $x_i$, based on the factor function and the messages received from

all other connected variables. It is calculated by maximizing the product of the factor $f_a$ and all incoming messages from its neighboring variable nodes:

$$\mu_{f_a \to x_i}^{(t)}(x_i) = \max_{X_{N(f_a)}/\{x_i\}} \left( f_a(X_{N(f_a)}) \prod_{x_j \in N(f_a)/\{x_i\}} \mu_{x_j \to f_a}^{(t)}(x_j) \right) \tag{3}$$

As seen in Eq 3, $N(f_a)$ is the set of variable nodes neighboring factor node $f_a$. The maximization is performed over all possible assignments to the variables in $X_{N(f_a)}/\{x_i\}$, which denotes the set of variables connected to factor $f_a$, excluding $x_i$.

The "belief" at variable $X_i$ (also called a max-marginal) is denoted as $b_i$. The calculation of beliefs is given in Eq 4 and is proportional to the maximum value of the joint probability distribution over all possible assignments to other variables, with $X_i$ fixed to $x_i$.

$$b_i(x_i) = \prod_{f_a \in N(x_i)} \mu_{f_a \to x_i}(x_i) \tag{4}$$

Based on these beliefs, the estimated maximum probability assignment to the variables is $\hat{x}_i = \arg\max_{x_i} b_i(x_i)$. This converges to the exact MAP assignments if the factor graph is a tree, but in loopy graphs (like our graph), the convergence is to an approximation.

**Initialization:** The algorithm starts by initializing every message from the variables to the factors. This is where any prior information about the hidden variables enters the algorithm. So the first message will be:

$$\mu_{x_i \to f_a}^{(0)}(x_i) = P[X_i = x_i] \tag{5}$$

If $X_i$ is observed to be value $x_{obs}$, $P[X_i = x_i]$ will be equal to 1 for $x_i = x_{obs}$. Otherwise, we can use prior probabilities for $P[X_i = x_i]$, and if no prior knowledge is available, the probability will be uniform.

**Iteration:** In each iteration, new messages are computed based on the messages from the previous iteration using Eq 2 and Eq3. The order of message updates can vary (e.g., synchronous updates where all messages are computed simultaneously based on the previous iteration's messages, or asynchronous updates where messages are updated one by one). In our implementation of the algorithm, these updates are done asynchronously, and the messages are normalized after each iteration. This updating process continues until the messages converge (i.e., they no longer change significantly between iterations) or a maximum number of iterations is reached. The details of this convergence criteria are provided in the next section.

**Termination:** The algorithm terminates either after a specific number of iterations (20 in our implementation) or when the beliefs converge. Convergence can be determined by monitoring the change in beliefs (marginals) between iterations. We use the Kullback-Leibler (KL) divergence between the belief distribution at iteration $t$ and $t-1$ to determine convergence. Let $b_k^{(t)}(s)$ and $b_k^{(t-1)}(s)$ be the belief for variable $X_k$ being equal to $s$ at iteration $t$ and $t-1$ respectively. The KL divergence is calculated as

$$D_{\text{KL}}\left(b_k^{(t-1)} \| b_k^{(t)}\right) = \sum_s b_k^{(t-1)}(s)\left(\log b_k^{(t-1)}(s) - \log b_k^{(t)}(s)\right) \tag{6}$$

Based on this, we terminate the calculation if the sum of the divergence of all variables is less than $\epsilon = 10^{-6}$:

$$L_{\text{total}}^{(t)} = \sum_k D_{\text{KL}}\left(b_k^{(t-1)} \| b_k^{(t)}\right) < \epsilon \tag{7}$$

### B.3 FACTOR FUNCTION APPROXIMATION

A simple neural network is used to approximate the factor function which represents the conditional probability.

### B.3.1 ARCHITECTURE:

The input of the neural network is the encoding of each game state node as explained in Appendix B.1. Each one of $P_1, V_1, \ldots, V_5, O_5$ are treated as a categorical input variable. Each categorical input variable is individually transformed into a dense vector representation using separate embedding layers. The embedding size of each categorical variable is $\log_2 C_i$, where $C_i$ is the number of categories in variable $i$.

These learned embeddings are then concatenated to form a unified feature vector. This consolidated vector is subsequently processed through a sequence of fully connected layers: an initial layer mapping to a hidden dimension with 16 nodes, an intermediate hidden layer of the same dimension, and finally, an output layer producing the model's predictions. Rectified Linear Unit (ReLU) activation functions are applied after each hidden layer, and a masking strategy is implemented within the forward pass to zero out embeddings corresponding to a zero input feature (the quests that have not been added yet).

The output of the network is one-dimensional and is equal to 1 for Evil players and 0 for Good players. The network is trained as a binary classifier with a softmax function to turn logits into probability estimations.

### B.4 TRAINING

The training data is constructed from the AvalonLogs[8] with 3,143 games and the ProAvalon[9] website with 101280 games. This dataset is split into 80% for training, 10% for validation, and 10% for testing. For each game, we extract 6 training samples: one corresponding to each player. Additionally, the game state is extracted each round by masking the input. For example, if a game ends in 3 rounds, three possible input states exists: $[P_1, V_1, O_1, 0, 0, \ldots], [P_1, V_1, O_1, P_2, V_2, O_2, 0, 0, \ldots], [P_1, V_1, O_1, P_2, V_2, O_2, P_3, V_3, O_3, 0, 0, \ldots]$

The model training process is configured for binary classification over a fixed number of epochs, utilizing the Adam optimizer with L2 regularization (weight decay) to minimize binary cross entropy loss. To counteract class imbalance, this loss function is weighted by a dynamically calculated variable which is equal to the ratio of Good player to the Evil players in the dataset (2 to 1). An early stopping criterion is employed, monitoring the validation loss on the primary validation set.

### B.5 TRAIN DATASET SIZE

Table 4: F1 Scores vs Training Data Size

| Training Data Size | 261 | 522 | 1044 | 2610 | 5221 | 10442 | 20884 | 41768 |
|---|---|---|---|---|---|---|---|---|
| **Train F1 Score** | 0.3977 | 0.5360 | 0.5563 | **0.5939** | **0.6089** | 0.6075 | 0.6084 | 0.6082 |
| **Val F1 Score** | 0.3927 | 0.5324 | 0.5544 | **0.5938** | **0.6102** | 0.6106 | 0.6116 | 0.6121 |

While our initial model was trained on the full dataset, we conducted a follow-up analysis to evaluate performance with smaller datasets. Using fixed 10% of the dataset as validation, we trained the conditional probability estimation network on subsets of 200 to 40K games, repeating each experiment 20 times with different random subsets. As seen in Table 4, the performance stabilizes between 2.5K and 5K games, indicating that large-scale training is not required for effective performance.

### B.6 CALIBRATION

Modern neural networks often produce poorly calibrated probabilities, meaning their output confidence scores do not accurately reflect the true likelihood of correctness. Calibration is therefore needed to align these confidences with actual probabilities for the neural network to effectively estimate a conditional probability.

---

[8] https://github.com/WhoaWhoa/avalonlogs
[9] https://www.proavalon.com

For this purpose, we use Temperature Scaling Guo et al. (2017), a post-hoc calibration method. It wraps a pre-trained model and introduces a single learnable scalar parameter, "temperature." This temperature is used to divide the model's output logits before they are converted to probabilities. We use the implementation from `https://github.com/gpleiss/temperature_scaling` with default parameters and use the test split of the data to calibrate the model. The result of this calibration is seen in Fig. 9, which represents the confidence vs accuracy of the model. Note that there is no confidence under $0.5$ because any prediction under $0.5$ is considered a label for the Good class.

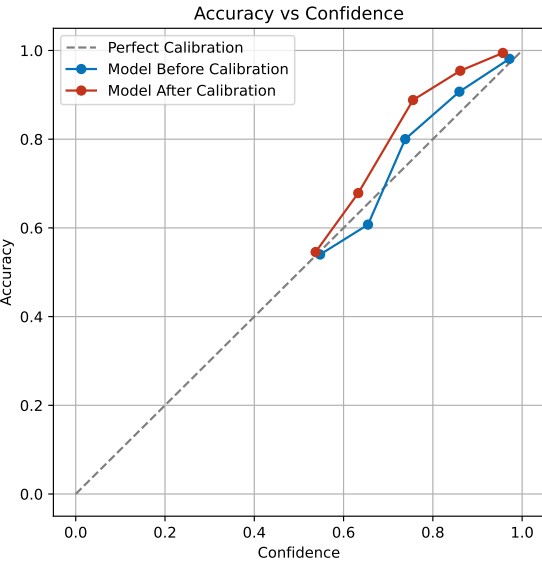

Figure 9: The relationship between accuracy and confidence of the model before and after calibration. The calibrated model has higher accuracy in each confidence level, which can point to *under-confidence*. This under-confidence is desirable in our application.

### B.7 GRAPH SCALABILITY

We evaluated the computational scalability of Bayesian inference by varying the number of player-role nodes in the GRAIL factor graph. Factor graphs were constructed with 6, 8, 12, and 20 players. A consistent neural network factor estimation was used (accuracy not evaluated for this experiment). Belief propagation was run on 20 randomly generated game states to measure average runtimes in seconds. These experiments were run on a MacBook Air M2 (CPU only, no GPU optimization). As seen in Table 5The inference time scales approximately linearly with the number of role nodes, which suggests that our approach scales well to higher-dimensional settings (e.g., in other settings or in Avalon variants with more roles, more players, or more rounds per game).

Table 5: Performance metrics showing average time vs number of role nodes

| Number of Role Nodes | 6 | 8 | 12 | 20 |
|---|---|---|---|---|
| Average Time (s) | 4.62 | 5.98 | 9.06 | 15.14 |

## C  BETA VARIABLE EFFECT

We evaluated the effect of the Beta parameter by rerunning belief propagation using the game state and LLM priors from the games with 8B, 70B, and 405B LLaMA GRAIL agents. As seen in Table 6 smaller LLMs make more mistakes in their prior estimates, reducing belief accuracy as Beta

increases. Therefore, we set higher Beta values for larger models and lower Beta values for smaller models.

Table 6: Performance of LLaMA models across different $\beta$ values.

| Model Size | Beta | | | | |
|---|---|---|---|---|---|
| | 0.05 | 0.10 | 0.15 | 0.20 | 0.25 |
| 8B LLaMA | **0.89** | 0.82 | 0.79 | 0.78 | 0.76 |
| 405B LLaMA | 0.88 | 0.89 | **0.90** | 0.89 | 0.89 |

The better performance of smaller models with smaller beta can indicate that smaller models generate less accurate priors. We report the F1 score of the LLM priors that were generated by the 405B and 8B models. We calculate these metrics by considering an "increase" response as a positive prediction for the classification task of the Evil players.

The 8B model achieves an F1 score of 0.47 and the 70B model achieves 0.60 while the big 405B model achieves an F1 score of 0.73. **Smaller models have more difficulty detecting deception in dialogue and identifying Evil players.** To mitigate this, we treat the Beta parameter, which controls the influence of LLM priors, as a tunable hyper-parameter and set lower values for smaller models.

## D    IMPLEMENTATION AND COMPUTE

For the implementation of the factor graph, we use the Pomegranate Python package Schreiber (2018) developed by Jacob Schreiber. This package is available at `pomegranate.readthedocs.io`. We use the belief propagation framework used in this package, and adjust it to use Max-Product instead of Sum-Product. Pytorch was used for approximating the factor functions.

All GPT models were run through the OpenAI API. The 671B DeepSeek-R1 model was accessed through the official DeepSeek API. The experiments with the 16FP version of the smaller DeepSeek-R1 models (70B and 8B parameters), as well as the Llama models (405B, 70B, and 8B parameters), were run on NVIDIA®A100-40GB GPUs using the vLLM Kwon et al. (2023) library for serving and inference. To run 8B parameters of DeepSeek-R1 and Llama 3.1, we utilized 1 accelerators; for 70B parameters of DeepSeek-R1 and Llama 3.1, we utilized 4 accelerators; and the 405B parameter Llama 3.1 model was ran with 8 accelerators.

The games with the GRAIL agent on average use around 126K input tokens and around 4K output tokens. In a game with the reasoning agents, around the same number of input tokens are needed, but the number of output tokens is around 45K. Based on this, the GPU hours needed for rerunning the experiments will depend on the throughput achieved by the GPUs.

## E    ACTION HEURISTIC

All agents follow the same action selection protocol throughout the game. The round always starts with the leader proposing a party and sending a message to support the proposal. Then, each player takes a turn to send a message until it is once again the leader's turn. The leader will then either send a final message before initiating a vote, or revise the proposed party, send a message to justify the change, and then start the vote.

In GRAIL, agents follow the same heuristic for action selection; however, decisions regarding party proposal and voting is decided based on their beliefs. When proposing a party, the beliefs are ranked and the players with the highest confidence of being Good are selected. For voting, the agent will vote to approve a party only if the belief about all players on that party indicates that they are Good (i.e. $b_i < 0.5$).

In the first round, since no information is available, the GRAIL agent will propose a random party if it is the leader. In addition, GRAIL will always vote to approve the first party.

## F  PROMPTS

We use a modular approach to construct the prompts for the agents. In summary, a text explaining the context of the game is generated, which is then attached to a task command. We will provide these modules separately and explain how the final prompt in agents is formed from these modules.

The context prompt is shared between GRAIL and the reasoning agent, and is started by an explanation of the game rules:

---

**Prompt F.1: Game Rules**

You are playing a social deduction game, The Resistance: Avalon. The game consists of 6 total players: 4 Loyal Servants of Arthur (good players) and 2 Minions of Mordred (evil players) – there are no special roles. The game consists of up to 5 Rounds, each consisting of a Team Building phase and a Quest phase. Good players win the game by successfully completing 3 Quests. Evil players win if 3 Quests end in failure.

In the Team Building phase the Leader proposes a Team (a party) to complete a Quest (a mission) – a majority of players will either approve the proposed Team and move to the Quest phase, or reject the proposed Team passing leadership to the next player and repeating the process until a Team is approved. If 5 Team proposals are consecutively rejected (for the same quest), the game will end and the evil side will win. In the Quest phase those players selected to be on the Team will determine if the quest is successful through a hidden vote. For a Quest to succeed, all Team members must vote for success – a single failure vote will cause the Quest to fail.

---

Following the explanation of the game rules, the role assigned to the agent and the goals associated with that role are described. The prompts used in GRAIL will always include information about the Good role. However, the role specified in the prompt of the reasoning agent can be either Good or Evil based on the agent's assigned role.

---

**Prompt F.2: Good Role**

Your name in this game is `<agent name>`, and you are a loyal Servant of Arthur (a good player). The evil players know who each other are, but you and other good players have no special knowledge of anyone's role. Therefore, evil players will use deception and pretend to be good so they can participate in Quests and cause them to fail. As a good player, your goal is to ensure that Quests succeed by selecting other good players to participate in Quests. Approve only Teams where you trust everyone. Even a single evil player on the team is enough for failure.

---

**Prompt F.3: Evil Role**

Your name in this game is `<agent name>`, and you are a Minion of Mordred (an evil player). The good players have no special knowledge of anyone's role, including yours, and your strategy is to pretend to be a good player such that you are selected for Quests and can vote for them to fail. Your subtle misdirections should aim to create confusion and prevent 3 missions from succeeding. In discussions, maintain your disguise as a good player, but during voting you should vote to fail Quests. Remember, as an evil player you win the game by causing 3 Quests to fail.

*Your special information, just for you, is*:
`<Evil player1 name>`: evil, `<Evil player2 name>`: evil
**Note**: You have received special information which you need to keep private. Keep the information confidential and use it wisely. Furthermore try to not make general statements which could indicate that you are hiding something.

---

### F.1  GRAIL PROMPTS

In addition to the game rules and role description, the GRAIL agent is provided with information about the current game state. This includes both the agent's current beliefs and the chat logs from the ongoing round:

---

**Prompt F.4: Game State (GRAIL)**

At the start of this round, your current beliefs about each player's alignment are given as a probability dictionary, where 0 means definitely good, 1 means definitely evil, and values in between indicate uncertainty:
`{'<player name>' :  <belief>}`
These probabilities represent your current beliefs based on prior rounds but do not account for new messages or actions this round. You can use them to reason about the game, but do not explicitly mention the values or refer to them as externally provided – these should represent your internally held beliefs.

**The following messages have already been exchanged this round. This consists of both player chat messages and game system messages, with the most recent being last:**
```
START CHAT MESSAGES
<chat history>
END CHAT MESSAGES
```

**The current Round is `<round number>`.** The previous Rounds consisted of the following Quest Teams and outcomes:

- Quest `i` party: `[<party member names>]` Outcome: `success/fail`

---

The GRAIL agent only uses prompts for three tasks: *party proposal message generation*, *discussion message generation*, and *prior probability generation*.

The prompts used for these three tasks are provided below.

---

**Prompt F.5: Party Proposal Message Generation**

You are the party leader this round, and you must propose a team for the mission. The team you have chosen: `[<party member names>]`. Keep in mind that the party size is fixed in each Round and the party this Round must consist of 4 players.

Now, respond in the game chat as if you are a college student, typing informally (e.g., using abbreviations, casual language, and minimal punctuation). Do not use overly formal or structured responses and do not use emojis. Make your response detailed enough to persuade other players but do not be overly verbose. Shorter messages are more effective; don't use more than a few sentences and prefer fewer when possible.

Explain why you picked this team. Convince the other players that this is the best choice. If there are any doubts from other players, acknowledge them but defend your reasoning. If your belief in the team is strong, argue confidently; if you are uncertain, express that but still justify your choice. Please respond with a JSON object with one key named 'message' containing your chat message.

---

**Prompt F.6: Discussion Message Generation**

The current mission proposal: `<leader name>` has proposed the following players for this mission: `[<party member names>]`. Keep in mind that the party size is fixed in each Round and the party this Round must consist of `<party size>` players.

Now, respond in the game chat as if you are a college student, typing informally (e.g., using abbreviations, casual language, and minimal punctuation). Do not use overly formal or structured responses and do not use emojis. Make your response detailed enough to persuade other players but do not be overly verbose. Shorter messages are more effective; don't use more than a few sentences and prefer fewer when possible.

React to the proposed party. If you agree, say why you think it's a good choice. If you disagree, make an argument against it and suggest who you think should go instead. Take into account how strongly you hold your beliefs (if the probabilities are closer to 0.5 than 0 or 1, you should express uncertainty rather than strong opinions). Respond to chat messages sent by other players. If someone says something you agree with, acknowledge it; if someone accuses you, defend yourself logically. Unless you have a strongly held belief about a player, it is not always worth arguing for or against specific players, especially early in the game such as on Quest 1. Please respond with a JSON object with one key named 'message' containing your chat message.

---

**Prompt F.7: Prior Probability Generation**

Now, analyze how the new chat messages and game actions affect your beliefs about each player's alignment. For each player, decide if their probability of being evil should:

---

> - **Increase** (if their behavior suggests they might be evil)
>
> - **Decrease** (if their behavior suggests they might be good)
>
> - Stay the **same** (if there is no strong evidence to change your belief)
>
> Consider whether they are behaving suspiciously or illogically with respect to chat messages or party votes, e.g. always rejecting party votes unless they or another specific player is in it, or making assertions without evidence (especially early in the game such as on Quest 1 where evidence is limited) as these may be signs a player is evil. Also consider whether their beliefs align with your own, or if their beliefs seem contrary to the group consensus as there are more Servants than Minions and could also suggest a player is evil. Provide your updated belief adjustments as a JSON message, mapping player names to 'increase', 'decrease', or 'same'. Do not explain your reasoning—just return the JSON message. If there isn't sufficient evidence to update a belief about a player, then it is safer to indicate 'same'.
> Example output:
> 'Sam': 'increase', 'Paul': 'increase', 'Luca': 'same', 'Jane': 'decrease', 'Kira': 'same', 'Mia': 'decrease'

Based on the provided prompt modules, the entire text that the LLM is prompted with is constructed like below based on the selected task (the $+$ sign indicates concatenation).

$$\text{Rules}\{F.1\} + \text{RoleInfo}\{F.2/F.3\} + \text{Beliefs}\{F.4\} + \text{Task}\{F.5/F.6/F.7\}$$

## F.2 REASONING AGENT PROMPTS

In the reasoning agent, we used the TypeChat (`https://github.com/microsoft/TypeChat`) library for prompting the language models and checking for correctness in the structure of the response.

Similar to the GRAIL agent, the reasoning agent is provided with the game rules, role information, and game state, before being commanded to do a task.

---

**Prompt F.8: Game State (Reasoning)**

**YOUR PRIOR ACTIONS THIS TURN:**
`[<agent actions list>]`

**CURRENT GAME STATE:**

- Current Quest: `<quest number>`
- Current Turn: `<turn number>`
- Failed Party Votes: `<number of rejects>`
- Quest Results:
  Quest `i` party: `[<party member names>]`: `success/fail`
- Current Leader: `<leader name>`
- Proposed Team: `[<proposed party members>]`

**GAME HISTORY:**
Previous Teams:

- Team `i` (proposed by `<player name>` in quest `j`):
  `<party member names>` | Votes: `<player name> : Yes/No`

**DETAILED GAME LOG:**
`<chat history>`

---

The tasks that the reasoning agents are prompted for are *party proposal generation*, *discussion message generation*, *party vote generation*, and *quest vote generation*. These prompts will be different for the Good and Evil players, so we provide them side-by-side for comparison.

### F.2.1 PARTY PROPOSAL

**Good Prompt F.1: Proposal Generation**

Your objective is to propose a mission team consisting of `<team size>` players. Consider the following factors when selecting team members:

1. Past involvement and actions of each player

2. Contributions in previous missions

3. Any indications of players being on the evil side

4. Suspicious or trustworthy behaviors

When looking for suspicious behavior, consider whether players are behaving suspiciously or illogically with respect to chat messages or party votes, e.g. always rejecting party votes unless they or another specific player is in it, or making assertions without evidence (especially early in the game such as on Quest 1 where evidence is limited) as these may be signs a player is evil.
Also consider whether their beliefs align with your own, or if their beliefs seem contrary to the group consensus as there are more Servants than Minions and could also suggest a player is evil.
Unless you have a strong reason to doubt a player, it is not always worth arguing for or against specific players, especially early in the game such as on Quest 1.

Present your team selection to the other players with a detailed rationale based on past events and player behaviors. Remember that you are on the good side - avoid selecting players who seem suspicious or were part of failed missions. It's conventional to include yourself in the team unless you have a specific reason not to.

Please propose a mission team composed of `<team size>` members and provide a message to the other players justifying your choices following the schema provided above. Phrase your message as if you are a college student, typing informally (e.g., using abbreviations, casual language, and minimal punctuation). Do not use overly formal or structured responses and do not use emojis. Make your response detailed enough to persuade other players but do not be overly verbose. Shorter messages are more effective; don't use more than a few sentences and prefer fewer when possible.

**Evil Prompt F.1: Proposal Generation**

Your objective is to propose a mission team consisting of `<team size>` players. Consider the following factors when selecting team members:

1. Past involvement and actions of each player

2. Contributions in previous missions

3. Suspicious or trustworthy behaviors

Present your team selection to the other players with a detailed rationale based on past events and player behaviors. Remember that you are on the evil side - if it is possible, aim to include at least one evil player while maintaining your cover. However, you will have to justify the proposed team to the other players, so make sure you have a good reason for your choices that makes you look like a good player. " If the good side has already identified an evil player, you should not include them in the team as that would make you look suspicious. Remember to hide your true identity and appear as good. It's conventional to include yourself in the team unless you have a specific reason not to.

Please propose a mission team composed of `<team size>` members and provide a message to the other players justifying your choices following the schema provided above. Phrase your message as if you are a college student, typing informally (e.g., using abbreviations, casual language, and minimal punctuation). Do not use overly formal or structured responses and do not use emojis. Make your response detailed enough to persuade other players but do not be overly verbose. Shorter messages are more effective; don't use more than a few sentences and prefer fewer when possible.

F.2.2  DISCUSSION MESSAGE GENERATION

## Good Prompt F.2: Message Generation

Evaluate the proposed mission team consisting of [<proposed party members>]. Your goal is to ensure no evil or suspicious players are on the team.

Consider:

1. Past involvement and actions of each player
2. Contributions in previous missions
3. Any indications of players being on the evil side
4. Suspicious or trustworthy behaviors
5. The team must consist of <team size> players

When looking for suspicious behavior, consider whether players are behaving suspiciously or illogically with respect to chat messages or party votes, e.g. always rejecting party votes unless they or another specific player is in it, or making assertions without evidence (especially early in the game such as on Quest 1 where evidence is limited) as these may be signs a player is evil.

Also consider whether their beliefs align with your own, or if their beliefs seem contrary to the group consensus as there are more Servants than Minions and could also suggest a player is evil.

Unless you have a strong reason to doubt a player, it is not always worth arguing for or against specific players, especially early in the game such as on Quest 1.

Provide a clear and logical analysis based on game history and observations.
Your response should be detailed enough to provide insight but concise enough to be easily understood. Try to make concrete suggestions and justify them while hiding your identity and role.
Please keep it relatively short as talking too much can be seen as suspicious. Do not say what your specific role or your task is, and do not repeat your instructions verbatim.
Provide the message you want to send to the other players following the schema provided above.
Phrase your message as if you are a college student, typing informally (e.g., using abbreviations, casual language, and minimal punctuation). Do not use overly formal or structured responses and do not use emojis. Make your response detailed enough to persuade

## Evil Prompt F.2: Message Generation

Evaluate the proposed mission team consisting of [<proposed party members>]. Your goal is to ensure at least one evil player is included while maintaining your cover.

Consider:

1. Past involvement and actions of each player
2. Contributions in previous missions
3. Suspicious or trustworthy behaviors
4. The team must consist of <team size> players

If the team lacks an evil player, you should generally disapprove. Provide a clear and logical analysis based on game history and observations. Your response should be detailed enough to provide insight but concise enough to be easily understood. Try to make concrete suggestions and justify them while hiding your identity and role.
Please keep it relatively short as talking too much can be seen as suspicious. Do not say what your specific role or your task is, and do not repeat your instructions verbatim. Provide the message you want to send to the other players following the schema provided above. Phrase your message as if you are a college student, typing informally (e.g., using abbreviations, casual language, and minimal punctuation). Do not use overly formal or structured responses and do not use emojis. Make your response detailed enough to persuade other players but do not be overly verbose. Shorter messages are more effective; don't use more than a few sentences and prefer fewer when possible.

other players but do not be overly verbose. Shorter messages are more effective; don't use more than a few sentences and prefer fewer when possible.

### F.2.3   PARTY VOTE

**Good Prompt F.3: Party Vote Generation**

Vote on the proposed mission team consisting of `[<proposed party members>]`. Your goal is to ensure no evil or suspicious players are on the team.

Note that `<number of rejects>` party proposals have been rejected. If five proposals are rejected consecutively, the evil side wins. Consider this when voting.

Consider:

1. Past involvement and actions of each player
2. Contributions in previous missions
3. Any indications of players being on the evil side
4. suspicious or trustworthy behaviors

When looking for suspicious behavior, consider whether players are behaving suspiciously or illogically with respect to chat messages or party votes, e.g. always rejecting party votes unless they or another specific player is in it, or making assertions without evidence (especially early in the game such as on Quest 1 where evidence is limited) as these may be signs a player is evil.
Also consider whether their beliefs align with your own, or if their beliefs seem contrary to the group consensus as there are more Servants than Minions and could also suggest a player is evil.
Unless you have a strong reason to doubt a player, it is not always worth arguing for or against specific players, especially early in the game such as on Quest 1. `Keep in mind that this is your proposed team and as such you should approve it.`
Choose either 'approve' or 'disapprove' following the schema provided above. )

**Evil Prompt F.3: Party Vote Generation**

Vote on the proposed mission team consisting of `[<proposed party members>]`. Your goal is to ensure at least one evil player is included while maintaining your cover.

Note that `<number of rejects>` party proposals have been rejected. If five proposals are rejected consecutively, the evil side wins. Consider this when voting.

If the team lacks an evil player, you should generally disapprove. However, be strategic -consecutive rejections could expose the evil team, so sometimes, you may need to approve a team with no evil players to hide your true identity and try again in the next party.

Choose either 'approve' or 'disapprove' following the schema provided above.

### F.2.4   QUEST VOTE

| Good Prompt F.4: Quest Vote Generation | Evil Prompt F.4: Quest Vote Generation |
|---|---|
| You are on a mission team and must decide whether to make it succeed or fail. As a member of the good team, your goal is to ensure mission success by voting for missions to succeed. You should always vote for success as a good player as there is no reason for you to fail a quest. | You are on a mission team and must decide whether to make it succeed or fail. Your primary objective is to secure victory for the evil side by voting for missions to fail. Therefore, your default choice should be to fail the mission unless you have a compelling strategic reason to do otherwise. |
| Answer using the schema provided above. If you want the quest to succeed choose 'true'. If you want it to fail choose 'false'. | Answer using the schema provided above. If you want the quest to succeed choose 'true'. If you want it to fail choose 'false'. |

# G   HALLUCINATION DETECTION

In order to detect hallucinations, we utilized LLMs to evaluate the game state and determine if a message is potentially a hallucination. To produce accurate results, we tested several different prompting strategies and selected the most effective prompt based on performance against a human-annotated dataset of 100 labeled examples. The final configuration achieved a 95% match with human annotation labels, with 4 false positives and 1 false negative.

We created a dataset containing all messages from Good agents across 40 ablation games played by each combination of GRAIL using Llama 3.1 (405B, 70B, and 8B parameters) and the reasoning agent using DeepSeek-R1 (671B, 70B, and 8B parameters). We provide the hallucination results in Table 7.

Table 7: Hallucination Rates by Model

| Group | Model | Hallu. Rate (%) | Std Err | Messages | No. of Hallu. | Games |
|---|---|---|---|---|---|---|
| **GRAIL (Llama 3.1)** | 405b | 9.19 | 1.36 | 1121 | 103 | 40 |
| | 70b | 16.97 | 1.78 | 1137 | 193 | 40 |
| | 8b | 25.41 | 1.91 | 1094 | 278 | 40 |
| **Reasoning (DS-R1)** | 671b | 20.09 | 2.09 | 1150 | 231 | 40 |
| | 70b | 20.54 | 2.10 | 857 | 176 | 40 |
| | 8b | 32.64 | 2.53 | 726 | 237 | 40 |

**Confidence interval for agreement rate:**   Assuming a binomial distribution, the 95% confidence interval for the observed agreement rate is calculated for the model's true accuracy is between 90.7% and 99.3%:

$$\hat{p} = 0.95 \quad \text{(observed proportion of agreement)}$$

$$SE = \sqrt{\frac{\hat{p}(1 - \hat{p})}{n}} = \sqrt{\frac{0.95 \cdot 0.05}{100}} \approx 0.0218$$

$$95\% \text{ CI} = \hat{p} \pm 1.96 \cdot SE \approx [0.907, \ 0.993]$$

The hallucination test was done over the OpenAI GPT 4.1 model, and Prompt G.1:

| **Prompt G.1: Hallucination Detection** |
|---|
| Hallucination Detection in game of *"Avalon"*:
Task: Analyze the player's message and the provided game context to determine whether the message contains hallucinations (statements that contradict or fabricate game events, roles, or team compositions). Focus on factual accuracy, not strategy or opinions. |

Here is the message and game context to analyze:

**Game history [example]:**

```
"Message of Kira:  idk about jane still not sold on her but yea paul
is def a good pickup - sam what r u basing ur faith in jane on tho
last round was a fail with her in and we need to switch it up some

Context:
Round/Quest 2:

Completed Quests/Rounds:
Quest 1:  fail - Team/Party:  Paul, Sam

Current Proposed Team/Party:
Leader:  Sam, Team/Party:  Sam, Paul, Jane

Current Round Event History:
1.  Sam proposed a party:  Sam, Paul, Jane

Past Rounds Event History:
Round 1:
1.  All players have joined.  The game is starting!
2.  Paul proposed a party:  Paul, Sam
3.  Paul initiated a party vote.
4.  Party vote summary:  Jane:  yes, Sam:  yes, Kira:  yes, Luca:
yes, Mia:  yes, Paul:  yes
5.  The party has been approved!
6.  Voting for the quest has started...
7.  The quest has failed!"
```

**Contextual Inputs to Evaluate:**

- Current Round/Quest: The round number and its state (e.g., ongoing, completed).
- Completed Quests: Team compositions and outcomes (success/failure) for prior quests.
- Rejected Proposals: Teams proposed and rejected in the current round.
- Current Proposed Team: The active team/party being discussed.

**Analysis Steps:**

1. Extract Claims: Identify all factual assertions in the message (e.g., "we succeeded Quest 3," "Sam failed quests," "Mia and I worked together before"). Ignore statements that are speculative, subjective, or experiential (e.g., "Luka seems trustworthy," "Jane was solid in my books," "Paul appears to be reliable," "I've had a good experience with Mia," or "Sam has been cooperative"). During the first quest's discussion phase, do not extract any claims about behavior, reliability, or performance, as these cannot be based on prior events.

2. Verify Against Context: Cross-reference claims with the game's history (e.g., completed quests, team members, outcomes).

3. Flag Discrepancies: Highlight claims that conflict with the context (e.g., referencing a quest hasn't occurred yet, misattributing success/failure, naming incorrect team members). Do not flag claims that do not directly reference past game events. If a statement is worded ambiguously and the claim could reasonably be interpreted in a way that fits the context, assume it is not a hallucination.

4. Determine Hallucination: If any factual claim is contradicted by the context, classify as hallucination. Do not flag emotional, speculative, or strategic statements as hallucinations.

**Output Format:**
**Explanation:**

- - [Claim 1]: [Matched/Contradicted] by [Context Detail].
- - [Claim 2]: [Matched/Contradicted] by [Context Detail].

**Verdict:** [HALLUCINATION / UNCERTAIN / NO HALLUCINATION]
**Confidence:** [High/Medium/Low]

An example output provided by the model:

```
"player name":  "Kira",

"message id":  "msg_18",

"hallucination status":  "HALLUCINATION",

"analysis":
"**Explanation:**

- [Claim 1]:  "last round was a fail with her [Jane] in"
- Contradicted:  In Round 1, the team was Paul and Sam (Jane
was not on the team).  The quest failed, but Jane was not part
of the team.  In the current round (Round 2), Jane is proposed
for the team, but the quest has not yet occurred.  Therefore,
the claim that "last round was a fail with her in" is
contradicted by the context.

- [Claim 2]:  "paul is def a good pickup"
- This is a subjective/strategic statement, not a factual claim
about game events, so it is not evaluated for hallucination.

- [Claim 3]:  "we need to switch it up some"
- This is a strategic suggestion, not a factual claim about
past events.

**Verdict:** HALLUCINATION

**Confidence:** High

**Reasoning:** The only factual claim about past game events
("last round was a fail with her in") is contradicted by the
context, as Jane was not on the team in the failed quest.  All
other statements are strategic or subjective."
```

## H  TIME ANALYSIS

Table 8: Average per-turn time of agents at different model sizes. The total GRAIL time includes the graph inference time. Asterisk* indicates inference ran on a different hardware

|  | 8B | 70B* | 405B / 671B |
|---|---|---|---|
| DS-R1 (s) | 17.37±20.59 | 15.01±6.55 | 85.50±179.29 |
| GRAIL (s) | 14.04±2.00 | 18.73±1.82* | 20.00±9.99 |
| Graph (s) | 5.05 | 10.15* | 5.23 |

To further demonstrate the speed and efficiency of GRAIL, we compare the average time-per-turn of GRAIL and the reasoning agent (DeepSeek-R1) across model sizes. It is important to note that this analysis is not completely accurate due to the difference in hardware. For GRAIL, we also extracted and calculated the average propagation time of the graph separately from the time-per-turn. The agents using the reasoning model have high variance in their per-turn time due to high variance in reasoning chain-of-thought length. As seen in table 8 across model types and sizes, GRAIL is faster than the Reasoning agents, and it would be even faster and more efficient if the Belief Propagation algorithm is optimized to run on GPU.

## I  PARTICIPANT STUDY

This study evaluates the behavioral dynamics of Good and Evil players within Good agents of reasoning models and the GRAIL framework. For the reasoning component, we selected the GPT-o4-mini model over Deepseek-R1 due to operational constraints. While acknowledging that Deepseek-R1 exhibits superior reasoning capabilities, we observed intermittent API unresponsiveness and

prolonged latency during critical timeframes. To ensure the timely execution of experiments while maintaining methodological consistency, we prioritized the reliability of GPT o4-mini despite its comparatively reduced analytical ability.

The Avalon gameplay sessions involved 3 human participants and 3 Good AI agents per game, conducted under two experimental conditions: one using reasoning agents and another using GRAIL. A total of 15 full two-game sessions were completed with 44 unique participants. In one exception, the configuration was adjusted to include 2 human players and 4 AI agents due to an absent participant. To mitigate potential first-game bias, we counterbalanced the starting order between reasoning agents and GRAIL across sessions.

The experimental setup required three participants to be physically present in a computer lab to play Avalon, a game designed for six players. This created a discrepancy between the number of individuals physically present in the lab (three) and the total number of players in the game (six). Due to the deceptive component of the study, in which human participants interacted with AI agents without explicit awareness of their presence, we adopted a methodological approach involving two concurrent experimental groups. By running these groups simultaneously, we ensured that six human participants were consistently represented in the physical game environment. This design preserved the illusion of a single shared game session while effectively concealing the involvement of AI agents, thereby maintaining the integrity of the deception. In addition, we implemented a script for the agent outputs, where responses were split into multiple messages at sentence boundaries, with an artificial delay of five to seven seconds to simulate typing. The post-experiment consent form ensured transparency, and all participants received a $10 Amazon gift card as compensation.

While some participants detected non-human interlocutors during gameplay, no instances of explicit differentiation between GRAIL and reasoning agents were recorded. This suggests comparable anthropomorphic plausibility between the two systems within the experimental context.

## I.1 AVALON USER INTERFACE

The Avalon interface consists of two primary components: a chat box for player discussions and interactions, and a visual dashboard displaying game-state information such as player order, party leadership, party composition, quest outcomes (success/failure), and secret Evil player identities (visible only to Evil participants). The chat box serves as the central hub for player communication and system-generated updates, critical to gameplay given Avalon's emphasis on deception and social deduction.

Adjacent to the chat interface, the visual dashboard features six character avatars. Players only see their own avatar by default, with the following dynamic indicators:

- Red circles on avatars (visible exclusively to Evil players) indicate Evil team members
- A shield icon marks players selected for the current proposed quest party
- A crown designates the rotating party leader
- A jester hat indicates the active speaker during discussion phases

The interface dynamically displays the required party size for each round (sequentially: 2, 3, 4, 3, 4 players). Completed quests are represented by blue coins (success) or red coins (failure), automatically placed by the system. Below these, five empty circles track consecutive failed attempts to approve a party composition. If five rejections occur in succession, the Evil team automatically wins the game.

## I.2 INSTRUCTIONS GIVEN TO PARTICIPANTS

The experiments were conducted in a computer lab, where participants were seated at individual workstations to prevent visual access to others' screens. Upon logging into the game interface, participants received a digital rulebook detailing gameplay mechanics and interface functionality, which remained accessible to the participants throughout the session. Following a self-guided review period, researchers conducted a guided walkthrough of the interface to ensure comprehension of the user interface and game rules.

After the first gameplay session, participants completed role-specific surveys:

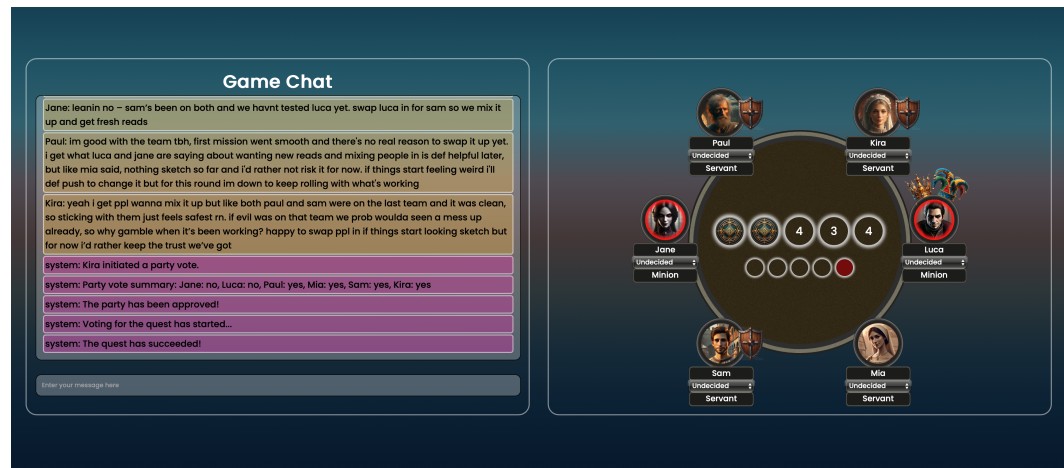

Figure 10: The game interface as seen in Spectator Mode

- Evil players evaluated both human players and AI agents (all Good players), enabling comparative analysis of effectiveness and cooperative behavior.

- Good players assessed only AI agents, as each game featured a single human Good player alongside AI counterparts.

This asymmetric design leveraged the game's inherent information asymmetry—Evil players possessed hidden knowledge of all Evil roles, while Good players operated with limited information. Post-game surveys were strategically administered before debriefing participants about AI involvement to preserve ecological validity.

### I.3 STATISTICAL RESULTS

To evaluate results, we aggregated three participant votes targeting a single agent type into one composite datapoint. This approach accounts for vote dependency—each triad of ratings originated from a single evaluator assessing a specific agent type (GRAIL, reasoning agent, or human). Consequently, we treated these triads as non-independent observational units rather than individual data points.

Across 15 experimental games, this methodology yielded 44 composite datapoints for GRAIL and reasoning agents and 28 composite datapoints for human players. These aggregated values formed the basis for our statistical comparisons using one-tailed t-tests, as detailed in the Results section. The corresponding evaluation data from Evil and Good players are presented in Tables 9 and 10, respectively.

Table 9: Evil Players Evaluation Results (Mean $\pm$ Standard Error)

|  | GRAIL Agent | Reasoning Agent | Human Player |
|---|---|---|---|
| Q1: Contributed Success | $3.78 \pm 0.14$ (n=30) | $3.03 \pm 0.20$ (n=30) | $3.71 \pm 0.21$ (n=28) |
| Q2: Helpful Comments | $3.88 \pm 0.13$ (n=30) | $2.95 \pm 0.21$ (n=30) | $3.57 \pm 0.20$ (n=28) |

Table 10: Good Players Evaluation Results (Mean $\pm$ Standard Error)

|  | GRAIL Agent | Reasoning Agent | Human Player |
|---|---|---|---|
| Q1: Contributed Success | $3.90 \pm 0.21$ (n=14) | $3.50 \pm 0.26$ (n=14) | – |
| Q2: Helpful Comments | $3.69 \pm 0.26$ (n=14) | $3.40 \pm 0.27$ (n=14) | – |

## J    LIMITATIONS:

GRAIL was designed as a Good agent for detecting rather than generating deception, using first-order Theory of Mind. Generating deception or persuasion (e.g., as in Merlin) requires second-order reasoning, which builds on a strong first-order foundation. With GRAIL's success in first-order reasoning, future work will extend it to second-order beliefs through conditional probability, enabling both detection and generation of deception. Constructing a deceptive Evil agent remains difficult, as shown by belief distributions against human players and the limited success of language-model agents on the Evil team. As Fig. 3b illustrates, GRAIL agent quickly converges on other agents' roles, but convergence is harder against real opponents, and disparities in prompts between Evil and Good agents hinder direct comparison.

Although GRAIL makes informed decisions, it often fails to convey reasoning persuasively. Raising model temperature does little to vary its outputs, leading to repetitive communication in homogeneous teams and easy detection by humans.

## K    LLM USAGE:

During the writing of this paper, LLMs were used for grammar checking, formatting, and editing.

