# OpenReview forum: "Bayesian Social Deduction with Graph-Informed Language Models"
_ICLR.cc/2026/Conference — ICLR 2026 Conference Withdrawn Submission_

### Official Review · Reviewer_k7rN · 2025-10-27

**Soundness:** 2
**Presentation:** 3
**Contribution:** 1
**Rating:** 4
**Confidence:** 4

**Summary:**

This paper introduces a neuro-symbolic method for a social deduction game. The proposed GRAIL utilizes a factor graph for Bayesian inference, estimate priors with language models, and propagate belief with factor graph. It augment language model outputs with the computed results.

The proposed model, with separately trained factor function approximation, outperforms other zero-shot reasoning model baselines, and surpasses human players in both winning rate and qualitative ratings.

Further analysis shows that the method uses fewer tokens overall. The ablation study demonstrates that both the language model and the graph components are essential for its success, with the graph determining the lower bound of performance.

Combining language models with probabilistic reasoning is an interesting direction. However, the results are not entirely significant, as the model is trained on additional data, and the experiments focused only on one simplified game.

**Strengths:**

- The overall presentation of the paper is clear and well-structured.
- The proposed integration of language models with probabilistic inference is interesting.
- The inclusion of both agent-agent and human evaluations makes the work more convincing. The experiments conducted in the Avalon domain are thorough.

**Weaknesses:**

- The use of a trained neural network for factor function approximation reduces the significance of the reported performance gains and makes the comparison with other methods potentially unfair.
- The experimental scope is limited. It remains unclear whether this approach generalizes to other domains or games. The trained neural network which is a key component of the system, needs to be specifically tailored and trained for each game rather than generalizable.
- The simplified Avalon setting further limits the significance of the results. As the authors noted, the original version includes different player roles (like Merlin). The GRAIL is also only evaluated as the Good players. This simplification may bias the game toward probabilistic methods rather than language-based social deduction (as also suggested by the ablation results, where the graph component is substantially more important than the language model in GRAIL). Evaluating the method in a more realistic setting could make the experiments more meaningful and the results more significant.

**Questions:**

- Is there a specific reason why the experiments were not conducted in a more realistic setting, such as evaluating GRAIL from both sides, and with different player roles? Also, could you clarify why including Merlin would 'introduce deception', as mentioned in the footnote on page 3?

- My understanding is that modeling other players’ beliefs corresponds to first-order Theory of Mind, whereas modeling an agent’s own beliefs does not. Since GRAIL acts as a player with full access to its own information, is the model in this work actually modeling its own belief? Could you elaborate this?

---

> ### Author Response · Authors · 2025-11-25
> **Authors Response part 1/2**
>
> We thank reviewer k7rN for their helpful and constructive review. We appreciate the acknowledgement of the strengths of our work, including the model's performance, the paper's presentation, and the thoroughness of the experiments. We provide our response in two parts due to character limit.
>
> ## Weaknesses:
> ### Trained Network:
> There are advantages and disadvantages imposed by any ML method (e.g., no free lunch theorem). The factor graph gives us computationally efficient and temporally consistent Bayesian inference, but this requires specifying conditional probabilities such as “given that a player was on a failed party, how likely are they to be Evil?” LLMs implicitly encode such assumptions, yet as our experiments show, these priors are often inaccurate, especially for smaller models. In principle, these conditional probabilities could be hand-crafted or estimated online from gameplay, but are likely to be brittle and error-prone. We chose to train a lightweight neural network on publicly available Avalon gameplay data to approximate these probabilities. Conceptually, this is no different from fine-tuning an LLM or other model on a task-specific dataset: it assumes access to data, but not to any information that competing methods could not also exploit (although in pilot studies, we found no improvement from fine-tuning LLMs with this gameplay data). Moreover, our data requirements are modest (see Appendix B.5). This design choice is what allows a 7B-parameter model in our framework to match the performance of much larger (≈600B) reasoning models.
>
>
> ### Generalization:
> We maintain that Avalon itself is an interesting and challenging environment and benchmark for assessing social deduction as a form of constrained probabilistic reasoning, as evidenced by prior work in this domain [1,2,3,4]. To reiterate, Avalon creates a rich environment for social reasoning in which the players must use a variety of social skills, such as deception, persuasion, teamwork, and compromise, to succeed. Furthermore, Avalon has a structured outline that introduces multiple constraints to the actions that players can take, and by making the future more predictable, it increases the importance of planning for both the good and evil teams. Also, it is important to note that "generalization to every game" is not a goal that all papers pursue. Our goal is to show what's possible and push SOTA forward, and analyze failures.
>
> At a high level, the GRAIL factor graph can be separated into three main components: Constraints, Roles, and Game State. In different games, we just need to change these components.
> The Game State depends entirely on the kind of information provided to the agent by the game engine. For example, in Mafia/Werewolf, the state can consist of events in each day/night cycle, such as votes and eliminations.
> Adapting Constraints and Roles for a new game type requires minimal information about the game. This can be done by answering a simple question: "What are the roles, and how many of each exist?"
> In variations of Avalon, adapting our method to other settings would require only minimal changes: updating the rules section of the prompt (Appendix E.1) and extending the factor graph by adding role nodes and adjusting the constraint factor node for team size. Factor functions would then be retrained using data from games with the new player configuration.
>
>
> ### Simplification:
> While it is true that we are using a more simplified version of Avalon, it is still noteworthy that LLM and LRM baselines fail to match humans' or GRAIL's performance even in this simplified version. The simplified environment still requires the good team to detect deceptive behavior and utilize persuasion when there is disagreement among players. In this sense, the simplified game still has the same characteristics that make Avalon a difficult challenge for language agents. And we emphasize that, despite a simplified version, the baselines are unable to play against humans and win. As we stated in the introduction, this demonstrates the shortcomings of LLMs in the area of constrained probabilistic reasoning. And it is also noteworthy that social reasoning always requires reasoning over uncertainty and constraints, regardless of the simplicity of the setting. A part of this reasoning in humans is done intuitively through system-1 thinking. But LLMs and LRMs try to do it explicitly in their CoT, which fails. So the factor graph, in a way, works as the implicit reasoner here.

---

> ### Author Response · Authors · 2025-11-25
> **Authors Response part 2/2**
>
> ## Questions:
>
> ### Experiments on Both Sides:
> GRAIL was designed for Good players to focus on detecting deception rather than generating it. Detecting deception requires first-order Theory of Mind, while generating deception or persuasion (like Merlin) involves second-order reasoning about others’ beliefs and intentions. For this work, we prioritized establishing a strong framework for first-order reasoning, as accurate second-order reasoning builds on it [5, 6]. Given GRAIL’s success in first-order reasoning, future work will extend the framework to estimate second-order beliefs using conditional probability, enabling agents to both detect and generate deception.
>
> The reason that having Merlin would introduce deception is that a part of Merlin's objectives is to hide his identity from the evil team. The Merlin player comes with the knowledge of which players are on which teams, so he can just disclose this information to other players. But in doing so, he reveals his identity to all players. The evil team has an option to try to assassinate whichever player they believe to be Merlin. If the actual Merlin is killed, the evil team would win. So this incentivises the Merlin player to engage in deceptive behavior.
>
> ### First order ToM:
> TL;DR: Modeling other players' beliefs about others corresponds to 2nd-order ToM, which is not modeled by GRAIL.
>
> We can define the hidden information about beliefs/mental states in an agent as $B=[b_0, b_1, b_2, ...]$ in which $b_i$ is the $i$th-order belief. In this case:
>
> * $b_o$ would be a player's beliefs about themselves (i.e., their role). This is the hidden information that each player has. This corresponds to $0$th-order ToM: assigning no beliefs or mental states to others
> * Then, $b_1$  would be a player's beliefs about the roles of other players. This corresponds to $1$st-order ToM: assuming that other players' actions are influenced by their roles. (This is what GRAIL models.)
> * $b_2$ corresponds to $2$nd-order ToM. That is a player's beliefs about the $1$st-order beliefs of others (e.g., "What I think other players believe about each other").
>
> In this framework, $b_1$ is the modeling (or inferring) of the $b_0$ of other players. Inferring $b_2$ in the agent requires the modeling of the $b_1$ of the other agents.
>
> ----
>
> ###### [1] J. Light, M. Cai, S. Shen, and Z. Hu, “AvalonBench: Evaluating LLMs Playing the Game of Avalon.” 2023. [Online]. ArXiv abs /2310.05036
> ###### [2] Simon Stepputtis, Joseph Campbell, Yaqi Xie, Zhengyang Qi, Wenxin Zhang, Ruiyi Wang, Sanketh Rangreji, Charles Lewis, and Katia Sycara. 2023. Long-Horizon Dialogue Understanding for Role Identification in the Game of Avalon with Large Language Models. In Findings of the Association for Computational Linguistics: EMNLP 2023, pages 11193–11208, Singapore. Association for Computational Linguistics.
> ###### [3] Yihuai Lan, Zhiqiang Hu, Lei Wang, Yang Wang, Deheng Ye, Peilin Zhao, Ee-Peng Lim, Hui Xiong, and Hao Wang. 2024. LLM-Based Agent Society Investigation: Collaboration and Confrontation in Avalon Gameplay. In Proceedings of the 2024 Conference on Empirical Methods in Natural Language Processing, pages 128–145, Miami, Florida, USA. Association for Computational Linguistics.
> ###### [4] Shi, Zijing, Meng Fang, Shunfeng Zheng, Shilong Deng, Ling Chen and Yali Du. “Cooperation on the Fly: Exploring Language Agents for Ad Hoc Teamwork in the Avalon Game.” ArXiv abs/2312.17515 (2023): n. pag.
>
>
> ###### [5] Braüner, T., Blackburn, P., & Polyanskaya, I. (2020). Being deceived: Information asymmetry in second-order false belief tasks. Topics in Cognitive Science, 12(2), 504–534.
>
> ###### [6] De Weerd, H., Verbrugge, R., & Verheij, B. (2015). Higher-order Theory of Mind in the Tacit Communication Game. Biologically Inspired Cognitive Architectures, 11, 10–21

---

### Official Review · Reviewer_SM4r · 2025-10-28

**Soundness:** 2
**Presentation:** 3
**Contribution:** 2
**Rating:** 2
**Confidence:** 4

**Summary:**

This paper presents GRAIL, a hybrid framework for social deduction games such as Avalon. The proposed model separates linguistic understanding and probabilistic reasoning: an LLM is used to interpret player dialogues and produce coarse-grained “directional priors”, while a factor graph performs Bayesian inference via max-product belief propagation to estimate hidden player roles.
Based on the inferred beliefs, the agent makes decisions such as party proposals and votes through a simple heuristic policy. The authors evaluate GRAIL against several reasoning-based and non-reasoning LLM baselines, showing improved win rates and greater stability, especially with smaller models.

**Strengths:**

1. The paper is clearly structured and easy to follow, with a clean separation of sections for modeling, inference, and experiments. The figures and appendices help convey the pipeline intuitively.

2. While the individual components (factor graphs, LLM priors, heuristic decision rules) are standard, their combination into a unified social deduction agent is a novel design choice. The idea of using LLMs only for qualitative “belief direction” is interesting and differs from typical chain-of-thought reasoning.

3. The method is technically sound, and the probabilistic formulation (max-product inference and neural factor approximation) is consistent. The experimental setup on Avalon is well-defined, and the human-agent evaluation adds credibility.

**Weaknesses:**

1. While the hybrid structure is elegant, I am not convinced that delegating the entire reasoning process to an external Bayesian graph is the right direction for long-term progress in social reasoning. Many reasoning chains in social deduction are inherently complex and multi-step, and the need for long chain-of-thought reasoning is not a weakness but a feature of such problems. By removing these steps from the LLM and handling them purely through a pre-defined probabilistic structure, the framework may gain stability but loses the very capacity for nuanced, emergent reasoning that LLMs are increasingly capable of developing. This design feels more like an engineering shortcut than a scalable reasoning principle.

2. The “higher / lower / same” directionality of language priors collapses the rich structure of social interaction into a single scalar. However, dialogue in social deduction games often contains entangled, relational cues (e.g., when one player’s statement implicitly reveals alliances or dependencies). These correlations cannot be expressed by independent prior adjustments on each player. In other words, the linguistic and relational signals are not separable; enforcing such separation risks discarding precisely the information that makes social reasoning interesting.

3. The paper linearly increases β to strengthen priors over rounds, but this schedule appears hand-tuned. It is unclear whether the same parameterization works across different LLMs, languages, or domains. Since β directly controls the interaction strength between the language layer and the factor graph, its sensitivity and robustness deserve deeper analysis—perhaps through a controlled ablation or sensitivity plot.

4. The paper primarily compares against older baselines (ReAct, DeepSeek-R1, etc.). However, recent works have advanced structured reasoning and dynamic workflow modeling for agentic systems—see AFlow (ICLR 2025), DyFlow (NeurIPS 2025), and MaAS (ICML 2025). These models would offer more competitive and conceptually relevant baselines. Without such comparisons, it is difficult to judge whether GRAIL’s improvements arise from genuine reasoning efficiency or simply from task-specific heuristics.

5. The framework assumes that LLMs can meaningfully answer prompts such as “is player X more suspicious than before?”, yet this capability is never validated. It is plausible that these judgments are inconsistent or even random. A simple perturbation test—flipping or shuffling a portion of these qualitative outputs—would reveal whether they actually carry semantic signal. Likewise, evaluating the consistency of priors across models (e.g., GPT-4 vs Llama-70B) could strengthen the empirical grounding.

**Questions:**

Most of my questions are already reflected in the Weaknesses section above, and I would appreciate detailed clarifications or additional experiments addressing those points.

Although I remain skeptical about several design choices, I am open to discussion and would be glad to reconsider my evaluation based on the authors’ responses and further evidence provided during the rebuttal phase.

---

> ### Author Response · Authors · 2025-11-25
> **Authors Response**
>
> We are grateful to reviewer SM4r for their time and effort in reviewing our submission. We thank the positive comments about the structure of the paper and the soundness of the method, and are encouraged by the novelty of our approach.
>
> ## Weaknesses:
> ### 1. Design decisions:
> It is arguable that chain-of-thought reasoning in LLMs is not true general-purpose reasoning, but rather, it is simply surface-level pattern matching [1,2,3]. These excessive, backtracking chains of thought are nowhere being efficient or optimal. This phenomenon is evident in the reasoning chains generated by the reasoning models when prompted with social reasoning questions: We observe that the DeepSeek R1 model generates long reasoning chains for more than 5-10 minutes for a trivial question. And even then, they struggle to accurately assess uncertainty [4,5,6] and maintain temporal consistency, something that is evident in our results and analysis as well. This is a fundamental problem of their architecture that cannot be addressed just by scaling (whether in training or at test-time). Tool-calling agentic frameworks and external reasoning are widely used [7,8], and in some sense, this is conceptually similar to our motivation. An LLM is not the ideal tool to do constrained probabilistic reasoning, but a PGM is.
>
> ### 2. The high/low/same scheme:
> We argue that the LLM is considering a holistic assessment: In light of all the dialogue and beliefs about other players, does the LLM think the current belief for a given player is accurate? It is important to note that the priors for each player are not adjusted independently of other players. Rather, all the priors are adjusted altogether.
>
> In the prompt in Appendix F.7, the LLM is being instructed to adjust the prediction, and we provide the game history to it as well. Prior adjustments are not independent of the already held beliefs about players and are used for further probabilistic reasoning. In other words, an adjustment of current beliefs will be used as priors for the next turn of the game.
>
> ### 3. Beta Parameter:
> We refer the reviewers to Table 3 and the "Effect of Beta" section in our submission, as well as Appendix C. We compared the changes in the F1 score of the beliefs between two model sizes based on the beta value. We ran the belief calculation using the same LLM judgment with different $\beta$ values. We see that a smaller model would get more accurate as the size of beta decreases, while a larger model has to rely on a relatively larger beta value to achieve the best results.
>
> ###  4. Baselines:
> At the time of experiments, DeepSeek R1 was (and still to some extent is) the state-of-the-art of open-source reasoning models. Many of the other baselines are unreasonable, as they weren't even released or are completely other types of agent frameworks designed for a different class of tasks.
>
> ### 5. LLM Judgment:
> As stated in lines 396, 397, and 398 in the “The effect of Beta” section (lines 400, 401, 402 in the revised version): We evaluate the LLM's judgment in generating the higher/lower/same judgments and report that the 405B llama model achieves an F1 score of 0.73 while smaller models achieve lower F1 scores. This shows that LLMs, if large enough, can in fact perform better on the “is player x suspicious than before” question. This shows that the LLM judgments are consistent and not random, and they scale with the size of the model.
>
> ---
> ###### [1] Parshin Shojaee et al. “The Illusion of Thinking: Understanding the Strengths and Limitations of Reasoning Models via the Lens of Problem Complexity.” ArXiv abs/2506.06941 (2025)
> ###### [2] Soumya Suvra Ghosal et al. “Does Thinking More always Help? Mirage of Test-Time Scaling in Reasoning Models.” (2025).
> ###### [3] Chengshuai Zhao et al. “Is Chain-of-Thought Reasoning of LLMs a Mirage? A Data Distribution Lens.” ArXiv abs/2508.01191 (2025)
> ###### [4] Kaitlyn Zhou et al. Navigating the Grey Area: How Expressions of Uncertainty and Overconfidence Affect Language Models. In Proceedings of the 2023 Conference on Empirical Methods in Natural Language Processing
> ###### [5] Kaitlyn Zhou et al. "Relying on the Unreliable: The Impact of Language Models’ Reluctance to Express Uncertainty". In Proceedings of the 62nd Annual Meeting of the Association for Computational Linguistics
> ###### [6] Mobina Pournemat et al. “Reasoning Under Uncertainty: Exploring Probabilistic Reasoning Capabilities of LLMs.” ArXiv abs/2509.10739 (2025)
> ###### [7] Long Hei Matthew Lam et al. "A Closer Look at Tool-based Logical Reasoning with LLMs: The Choice of Tool Matters". In Proceedings of the 22nd Annual Workshop of the Australasian Language Technology Association
> ###### [8] Theo Olausson et al. "LINC: A Neurosymbolic Approach for Logical Reasoning by Combining Language Models with First-Order Logic Provers." In Proceedings of the 2023 Conference on Empirical Methods in Natural Language Processing

---

### Official Review · Reviewer_EHJ9 · 2025-11-03

**Soundness:** 2
**Presentation:** 2
**Contribution:** 2
**Rating:** 4
**Confidence:** 3

**Summary:**

This paper uses the social deduction game Avalon to evaluate and improve LLMs' social reasoning abilities. The authors introduce a hybrid probabilisitic reasoning framework called GRAIL, which achieves competitive performance compared to strong reasoning models and can defeat human players in a controlled study. They also perform thorough analysis on the method with different model families and sizes, allowing the reader to more deeply understand the strengths and limitations of the method.

**Strengths:**

- Social reasoning is an important topic in AI and LLM research, which this work engages with.
- The proposed method is principled and performs well.
- This work conducts many kinds of analysis, including on resource usage and hallucination. It tests the method using different models (e.g., Llama and DeepSeek, varying sizes).
- This work conducts model vs human studies, supporting the effectiveness of the proposed method.

**Weaknesses:**

- This work only studies one game: Avalon. While it is not necessary here to extend GRAIL to other domains, I'd appreciate if the authors include more discussions on where they think GRAIL can also apply to (e.g., other games or social reasoning settings) and where it would face challenges.
- Why is ReCon an appropriate baseline and in fact the only non-reasoning-model baseline? The authors need to introduce ReCon more (given its current role) and argue why it makes sense here.

**Questions:**

How is this study different from the previous studies that evaluate Avalon gameplay with LLMs, and how is GRAIL different from previous work that applies probabilistic graphical models to social deduction games? The authors do cite relevant work, but have not explicitly articulated what they consider to be significant or novel compared to prior work. This is important for clear paper writing. I would raise my score if the authors address this point.

---

> ### Author Response · Authors · 2025-11-25
> **Authors Response**
>
> We thank reviewer EHJ9 for their thoughtful and comprehensive review. We appreciate the acknowledgement of the strengths of our work, such as the importance of the topic, the principled method, and the scope of our analysis. We are encouraged by the comments.
>
> ## Weaknesses:
> ### This work only studies one game:
> * *How can we extend the GRAIL framework to other games?*
> At a high level, the GRAIL factor graph can be separated into three main components: Constraints, Roles, and Game State. In different games, we just need to change these components.
> The Game State depends entirely on the kind of information provided to the agent by the game engine. For example, in Mafia/Werewolf, the state can consist of events in each day/night cycle, such as votes and eliminations.
> Adapting Constraints and Roles for a new game type requires minimal information about the game. This can be done by answering a simple question: "What are the roles, and how many of each exist?"
> In variations of Avalon, adapting our method to other settings would require only minimal changes: updating the rules section of the prompt (Appendix E.1) and extending the factor graph by adding role nodes and adjusting the constraint factor node for team size. Factor functions would then be retrained using data from games with the new player configuration.
>
> * *Where can GRAIL be applied?*
> Generally, any setting requiring constrained probabilistic reasoning can benefit from incorporating factor graph inference with an LLM. This includes social deduction, but it is not limited to it. For example, the factor graph can be used for Theory-of-Mind inference in structured environments.
>
>
> ### Recon as a Baseline
>
> * *Why is ReCon the only baseline?* ReCon, like GRAIL, is an open-source method that is independent of the underlying LLM model being used. So it can be compared effectively. We acknowledge the need for further introduction of the ReCon method. We have added an introduction to ReCon in section 5 of the revised paper that you can access.
> Older methods, such as [1], lack the modality of language, which makes them unfit for comparison among language-based agents. The recent LLM-based works, such as [2], are mostly benchmark-focused and do not focus on developing effective agents for Avalon.
>
>
> ## Questions:
> * *How is this study different?*
> This is **the first study** that evaluates the agents against **human players** with dialogue, and our agent is able to win against humans, and is scored higher than the baselines by humans. We are the first ones to test **reasoning models** in social deduction games; we are the first study to comprehensively study the **model size, hallucination, and test-time efficiency** of these models in social deduction settings. We create a framework for constrained probabilistic reasoning that is scalable and efficient.
> * *How is GRAIL different from previous work that applies PGMs?*
> It is important to note that [3] is, to the best of our knowledge *the only* prior work that mentions PGMs. Furthermore, [3] does not explicitly model the game variables within a PGM. In their method, the probabilities are not learned or modelled, and the inference on the PGM is not done using rigorous Bayesian inference, but rather, the PGM is just used as a way to decide what kind of context should be provided to the LLM when it is being prompted. Furthermore, they test their work on simple, hidden-word social deduction games (where one team must deduce the hidden word of the other team from clues), and it has minimal deception compared to the richer, multi-turn, and long-horizon environment of Avalon.
>
> ----
>
> ###### [1] Jack Serrino, Max Kleiman-Weiner, David C. Parkes, and Joshua B. Tenenbaum. 2019. Finding friend and foe in multi-agent games. Proceedings of the 33rd International Conference on Neural Information Processing Systems. Curran Associates Inc., Red Hook, NY, USA, Article 113, 1251–1261.
>
> ###### [2] Jonathan Light, Min Cai, Sheng Shen, and Ziniu Hu. 2023. Avalonbench: Evaluating llms playing the game of avalon. In NeurIPS 2023 Foundation Models for Decision Making Workshop
> ###### [3] Lin Xu, Zhiyuan Hu, Daquan Zhou, Hongyu Ren, Zhen Dong, Kurt Keutzer, See-Kiong Ng, and Jiashi Feng. 2024. MAgIC: Investigation of Large Language Model Powered Multi-Agent in Cognition, Adaptability, Rationality and Collaboration. In Proceedings of the 2024 Conference on Empirical Methods in Natural Language Processing, pages 7315–7332, Miami, Florida, USA. Association for Computational Linguistics.

---

### Author Response · Authors · 2025-12-04
**Summary and Final Remarks**

Dear Area and Program Chairs



Due to the unforeseen circumstances that have affected the process of reviews, we are providing a summary our paper, the reviews, and our responses.



We thank the reviewers for their detailed reviews and constructive, thoughtful comments during the discussion period. We appreciate their time and effort for helping us improve our submission.


## Paper Summary:
In this paper, we have introduced GRAIL, a neurosymbolic hybrid method for constrained probabilistic reasoning with LLMs that we have applied to the social deduction game Avalon. GRAIL surpasses the performance of large reasoning models, with a smaller, non-reasoning model. GRAIL is more efficient than the baselines both in token count and wall-clock time. Through ablation studies, we showed the importance of different components of GRAIL. Detailed analysis shows that GRAIL is more consistent and grounded in the game state, and makes fewer reasoning mistakes.

GRAIL is the first language agent that has been tested against humans in the social deduction game of Avalon, and is able to achieve a winrate of %67, surpassing the baselines both in quantitative and qualitative metrics.


## Reviews:
The reviewers have noted the novelty of our neurosymbolic principled approach and have praised the technical soundness and consistency of the method.

The reviewers have mentioned the relevance and importance of social reasoning. The comprehensiveness of our empirical analysis and the scope of experiments, as well as the human study, was unanimously complimented.



The main concerns brought up by the reviewers, and our response are as follows:



* **Baseline Choice**: At the time of experiments, DeepSeek R1 was (and still to some extent is) the state-of-the-art of open-source reasoning models. The ReCon agent was selected as the non-reasoning baseline because it is an open-source method that is independent of the underlying LLM model being used. We have added a section to the paper to introduce the ReCon method. Many of the other baselines are unreasonable, as they weren't even released or are completely other types of agent frameworks designed for a different class of tasks.

* **Extending GRAIL**: GRAIL can be applied to different settings by altering anyone of its three components: Constraints, Roles, and Game State. We also provide evidence that Avalon by itself is an interesting and challenging task for LLMs that requires a variety of skills and capabilities that still have not emerged in language models. Furthermore, we demonstrate that even a simplified Avalon is sufficient as a testbed because SoTA LLMs fail to match human performance.

* **Design Decisions**: We believe our analysis demonstrates the shortcomings of the Chain-of-Thought based scaling for social reasoning. Explicit social reasoning requires reasoning about uncertainty under constraints. LLMs cannot perform such reasoning, and our Factor Graph-based method addresses this issue.

---

### Note · Authors · 2026-01-06

I have read and agree with the venue's withdrawal policy on behalf of myself and my co-authors.